# CAUSAL REPRESENTATION MEETS STOCHASTIC MODELING UNDER GENERIC GEOMETRY

## ABSTRACT

Learning meaningful causal representations from observations has emerged as a crucial task for facilitating machine learning applications and driving scientific discoveries in fields such as climate science, biology, and physics. This process involves disentangling high-level latent variables and their causal relationships from low-level observations. Previous work in this area that achieves identifiability typically focuses on cases where the observations are either i.i.d. or follow a latent discrete-time process. Nevertheless, many real-world settings require the identification of latent variables that are stochastic processes (e.g., a multivariate point process). To this end, we develop identifiable causal representation learning for continuous-time latent stochastic point processes. We study the theoretical identifiability by analyzing the geometry of the parameter space. Furthermore, based on this, we develop MUTATE, a variational autoencoder framework with a time-adaptive transition module to evaluate stochastic dynamics. Across simulated and empirical studies, we find that MUTATE has the potential to answer questions in numerous scientific fields.

## 1 INTRODUCTION

Inferring causal relationships among variables from observations capitalizes the potential of machine learning to advance scientific discovery, as it reveals underlying mechanisms that are not identifiable from observational distributions alone (Pearl, 2009). However, we often do not have access to the causal variables but only the high-dimensional perceptual data, and causal variables with their structures are unknown and thus need to be learned. Yet, these latent causal variables are often not identifiable (Hyvärinen & Pajunen, 1999; Sorrenson et al., 2020). Recently, a growing number of studies on the disentanglement of latent causal representations have developed identifiability guarantees and proposed methods for estimating latent causal variables. Seminal works among them establish identifiability by leveraging sufficient variability in latent distribution arising from multiple-source data (Yao et al., 2022a; Song et al., 2023), auxiliary variable (Hyvärinen & Pajunen, 1999; Hyvarinen & Morioka, 2016; 2017; Hyvarinen et al., 2019), or intervention to a latent causal graph (Ahuja et al., 2023; Squires et al., 2023; Jiang & Aragam, 2023; Bing et al., 2024; Buchholz et al., 2023).

Most recent work mentioned above aims to recover the latent causal variables that follow a discrete-time process (Yao et al., 2022a; Song et al., 2023) and that are mixed by an invertible function. However, many latent causal variables of interest are continuous-time processes in practice; and the study of latent continuous-time causal variables driven by stochastic processes or systems of stochastic differential equations has received little attention, especially when mixing functions are non-invertible and more generic[1] For example, in video surveillance systems, cameras are strategically placed to detect and deter crime, safeguard against potential

---

[1]Following the standard usage in algebraic geometry, "a generic point of $X$ has property $P$" means that there exists a dense open subset $U \subseteq X$ such that every point of $U$ has property $P$ (Eisenbud & Harris, 2000, Ch. I).

threats to the public, and manage emergency response situations during natural and man-made disasters (Lima et al., 2020; Bacry & Muzy, 2014b; Bacry et al., 2012). In biology, fatal diseases such as cancer are principally caused by multiple cumulative mutations in driver genes as the colonial expansion proceeds. In neuroscience, the latent event dynamics trigger visible biological signals (Reynaud-Bouret & Schbath, 2010; Lorch et al., 2024). Finding cancer-associated mutational genes and tracking their behavior through their representation has been given much more paramount importance in recent few decades (Torkamani & Schork, 2009; Bailey et al., 2018; Nourbakhsh et al., 2024). Driven by the practical promise across applications, we study *when continuous-time latent stochastic point processes and their causal structure are identifiable*, and develop *algorithms to learn these latent dynamics from high-dimensional data*.

**Contributions.**

- We establish the first necessary and sufficient conditions that guarantee the full identifiability of latent point processes under generic, non-invertible mixing.
- We propose MUTATE, a novel identifiable variational auto-encoding method for learning causal representations of stochastic point processes.

## 2 PROBLEM SETUP: CAUSAL REPRESENTATION WITH STOCHASTIC POINT PROCESS

### 2.1 PRELIMINARIES AND NOTATIONS

Let $O_t \in \mathbb{R}^n$ be observable data, and $Z_t \in \mathbb{R}^p$ be a latent causal process with independent noise $\epsilon_t \in \mathbb{R}^p$. $O_t$ is being generated from latent point processes $Z_t$ through an unknown, arbitrary mixing function $f$. A multi-way array $A^{\otimes d}$ denotes the tensor/Kronecker product. In a time process, $\Phi \in \mathbb{R}^{p \times p}$ denotes the transition operator (e.g., an autoregressive coefficient matrix or continuous kernel matrix) and the symbol $\star$ represents the convolution operator with kernel effects. We assume a probability space $(S, \mathcal{B}(S), \mathbb{P})$, where $S$ is a Polish space (i.e., a complete separable metric space), $\mathcal{B}(S)$ is the Borel $\sigma$-algebra, and $\mathbb{P}$ is the probability measure, with $\mu$ a generic measure (e.g., for noise or intensity). $\mathcal{F}_t$ is the natural filtration up to the time $t$ of a process. Let $K$ denote an algebraically closed field[2] of characteristic zero. Throughout, and unless specified otherwise, we work over this field $K$.

### 2.2 A GENERATIVE MODEL FOR STOCHASTIC POINT PROCESSES

Throughout this paper, we consider a branch of non-homogeneous stochastic processes (Hawkes process) with dynamics governed by a conditional intensity defined as follows.

**Definition 2.1** (Conditional intensity, informal (Bacry & Muzy, 2014a))**.** Suppose a collection of latent processes that evolve stochastically and exhibit self-exciting dynamics over time. Specifically, let $Z_t := N_t$ denote the cumulative count process up to time $t$. We write $i \leftarrow j$ to indicate that the process $j$ exerts an influence on $i$. Accordingly, the conditional intensity of process $i$ at time $t$ is given by

$$\lambda_t^i = \mu_i + \sum_j \int_0^t \phi_{i \leftarrow j}(t - s) \, N_s(\Delta)^j,$$

where $\mu_i \in U$ is the baseline rate and $\phi_{i \leftarrow j} \in \Phi$ characterizes the excitation kernel from process $j$ to $i$. The counting process $N_t^i$ and the conditional intensity $\lambda_t^i$ satisfy: $N_{t+\Delta}^i - N_t^i = N_t(\Delta)^i$ and $\lambda_t^i = \frac{\mathbb{E}[dN_t^i | \mathcal{F}_t]}{dt}$.

---

[2]By definition, a field $k$ is *algebraically closed* if every non-constant polynomial $f(x) \in k[x]$ has a root in $k$, or equivalently, if $k$ admits no proper algebraic extension (Lang, 2002; Atiyah & Macdonald, 1969, Ch. V).

For such a point process to be well-defined, some non-trivial constraints are required, one of which is the stationary condition, an assumption widely adopted in most stochastic process literature to ensure the uniqueness of the process.

**Assumption 1.** *1. (Stationary increments) The process $N_t^{(\Delta)}$ is wide-sense stationary, i.e., its first and second moments exist and are time-invariant. In particular, the intensity process $\mathbb{E}[\Lambda_t]$ is uniformly bounded and $dN_t^{(\Delta)}$ has stationary increments.*

*2. (Kernel Integrability) The convolutional causal kernel $\Phi_t \in \mathbb{R}^{p \times p}$ is square-integrable, i.e.,*

$$\int_0^\infty \|\Phi_t\|_F^2 \, dt < \infty,$$

*where $\|\cdot\|_F$ denotes the Frobenius norm.*

Now, we formally set up the problem of identifying the generative model of stochastic point processes. We consider a collection of unstructured low-level observations $O = (O_t)_{t \leq T}$ generated from the latent process $N_t$ through an arbitrary mixing function $f$. Compactly, by absorbing the kernel matrix $\Phi$ and the counting process $N_t$ into a standard convolution operator, the generative model can be written as

$$O_t = f(N_t(\Delta)), \qquad \lambda_t = U + \Phi_t \star N_t(\Delta). \tag{1}$$

Thus, the central goal is to recover the parameter space $\Theta := (f, N_t, \lambda_t, \Phi, U)$ given samples or the full distribution of observations $O_t$. Concerning the theoretical soundness, we adopt the setting where the form of the mixing function, the number of latent causal processes, and their causal structure are fully unknown.

## 3 IDENTIFIABILITY THEORY

In this section, we establish the identifiability of the latent causal stochastic point process. We begin by introducing a family of general equivalent classes, a model that can be maximally identified from the given data. Then, the geometry of the parameter space of this equivalent class is examined to ensure the full recovery of both the mixing map and kernels, together with causal structure through the algebraic structure of the parameters. All detailed proofs are deferred to Appendix B and C, and the discussion on generalization of our identifiability can be found in Appendix D.

### 3.1 MAXIMALLY IDENTIFIABLE EQUIVALENT CLASSES

We begin by introducing the maximal equivalent class that can be identified from discrete-time observations. Suppose we observe a discrete-time observation sequence $O_{t_0}, O_{t_0+\Delta}, O_{t_0+2\Delta}, \ldots, O_{t_0+k\Delta}$ at times $t_0, t_0 + \Delta, t_0 + 2\Delta, \ldots, t_0 + k\Delta$. Given a linear Hawkes-type intensity, we are provided a discretized latent process $Z_t^{(\Delta)}$ under the subsequence $\Delta$, with its associated intensity: $\lambda_t^{(\Delta)} = u + \phi(\Delta) \cdot \Delta dN_t^{(\Delta)}$. The discrepancy between realizations arises due to the mismatch between the continuous-time dynamics and its discrete approximation, i.e., $\lambda_t^{(\Delta \to 0)} \neq \lambda_t^{(\Delta)}$, which implies that only latent processes generated under the same discretization scale $\Delta$ as the observation resolution can be recovered from $O_t^{(\Delta)}$. Therefore, the identifiability of the underlying latent dynamics is constrained to a discrete-time equivalence class determined by the resolution of observation. To capture the distribution-level changes and dynamics, we argue that recovering the distribution behavior of the latents suffices in most scientific tasks, and it can be used to generate the latents of any other $\Delta$ scale. Accordingly, we are able to identify only an equivalence class, as introduced in the subsequent definition.

**Definition 3.1** (Weakly-convergent equivalent class). Let $(dN_t, \lambda_t)$ denote the ground-truth latent point process and its associated continuous-time intensity function. A pair $(Z^{(\Delta)}, \lambda^{(\Delta)})$ is said to belong to the

**Weakly-convergent equivalent class** of $(dN_t, \lambda_t)$ if it satisfies the following weak convergence condition:

$$(Z^{(\Delta)}, \lambda^{(\Delta)}) \xrightarrow[\Delta \to 0]{d} (dN_t, \lambda_t),$$

i.e., the estimated latent process and its discrete-time intensity converge in distribution to the ground-truth continuous-time process as the resolution parameter $\Delta \to 0$.

Thanks to Definition 3.1, it is sufficient to find such a model belonging to the equivalent class and establish its identifiability. Following Kirchner (2016), we revisit the close connection between order-$l$ integer-value autoregressive processes (INAR($l$)) and the multivariate stochastic point process through convergence limits when $l \to \infty$. An INAR($\infty$) process is defined as the infinite-order autoregressive model with integer variables $Z_t$. In particular, it has the form:

$$Z_t = \sum_{\tau > 0}^{\infty} a_\tau Z_{t-\tau} + \epsilon_t, \tag{2}$$

where $a_\tau$ is a constant coefficient and $\epsilon$ is a mutually time-wise independent noise. Our intuition is that replacing the constant coefficient with a time-invariant kernel $\phi_t$ still ensures the weak convergence to a stochastic point process $N_t$. The main goal is to show that such a replacement is a member of the defined weakly-convergent class, as given in the following lemmas.

**Lemma 1** (Bounding point process in Variational approximation). *Let $N_t \in \mathbb{R}^p$ be a multivariate point process whose conditional intensity function $\lambda_t$ is governed by a convolution structure described in Eq. (1) and $\epsilon_t$ is a mean-zero and mutually independent noise. Then the intensity model admits the following weak convergence:*

$$Z_k^\Delta := \lambda_k^\Delta + \epsilon_k^\Delta \xrightarrow{w} N_k^\Delta, \tag{3}$$

*where the subscript $k$ denotes an arbitrary subsequence process and $\lambda_k^\Delta$ is the corresponding intensity under the same subsequence.*

**Lemma 2** (Convergence to latent equivalent classes). *Assume the weak convergence condition in Lemma 1 is satisfied. Then, there exists a latent process $Z_t^\Delta$ such that the process $N_t$ and its variational approximation converge in distribution to the same latent causal class. Formally,*

$$\lim_{\Delta \downarrow 0} N_k^\Delta := Z^\Delta \xrightarrow{d} N_t. \tag{4}$$

*This implies that, up to infinitesimal resolution $\Delta$, the estimated process admits a latent representation governed by the same convolution dynamics.*

Lemma 1 and 2 together establish the weak convergence of continuous stochastic processes under the corresponding weak topology. Roughly, for any compact time interval $[a, b]$, a subsequence process of the original process under such an interval converges to a *continuous-time* causal point process $N_t$.

This convergence ensures that $Z^\Delta$ effectively represents $N_t$ and maintains all causal structures. Without loss of generality, we can therefore directly work with $Z^\Delta$ and study its identifiability by analyzing the geometry of the associated parameter space.

### 3.2 Geometry Characterization of Model Identifiability

As a stepping stone toward our main results, the geometry of the proposed latent model characterizes the uniqueness of the parameters in the generative model. An exploration by Carreno et al. (2024) has shown that parameter space can be fully recovered if and only if the solution set of the system defined by the available data is zero-dimensional, i.e., it consists of finitely many points consistent with the number of latent variables

or parameters. Geometrically, given the finite-dimensional observation distribution $P(O_t)$, we consider the ideal

$$\mathcal{I} = \langle P(O_t) - P_\Theta \rangle, \tag{5}$$

and identifiability of the parameter $\Theta$ requires that $\mathcal{I}$ has dimension zero. Intuitively, the parameter space $\Theta := (f, Z^\Delta)$ of the generative model (i.e., the mixing map $f$ together with the full parameters of $Z^\Delta$) corresponds to the vanishing locus of $\mathcal{I}$, which in this case cannot lie on any higher-dimensional hypersurface, as shown in Figure 1. In practice, the full distribution is typically inaccessible; hence, we aim to establish identifiability by relying solely on partial distributional information. Following Wang & Seigal (2024), we choose the cumulant of the observational distribution as an intermediary to study the geometry of parameter space $\Theta$. Cumulant of infinite order is an important algebro-geometric signature, as it precisely encodes the entire distribution, including the component-wise and time-wise dependency among variables (Achab et al., 2018; Jovanović et al., 2015; Landsberg, 2011). Higher-order cumulants then capture the causal structure of a distribution at the same orders, enabling fine-grained mathematical analysis of intervention effects beyond traditional mean and variance shifts.

Under generic (non-Gaussian) conditions, Carreno et al. (2024) show that if the observed variable satisfies a fully linear causal model of the form $X = FZ$ where $Z = AZ + \epsilon$, then the $d$-th cumulant of $X$, denoted by $\kappa_d(X)$, admits a unique decomposition as

$$\kappa_d(X) = \sum_{j=1}^{p} \kappa_d(\epsilon) \cdot K_j \otimes \cdots \otimes K_j, \quad K = F(\mathbb{I} - A)^{-1}.$$

Compared to Eq. (5), the above equation induces a simplified ideal $\mathcal{I} := \langle K(\mathbb{I} - A) - F \rangle$, with $A$ and $F$ treated as generic indeterminates. Consequently, the parameter space encoded in $K$ is identifiable up to scaling and permutation. The identifiability of this linear mixture of parameters is equivalent to showing that the algebraic variety defined by $\kappa_d(X)$ is *not* $p$-defective (Chiantini et al., 2017). This connection between the geometry of the parameters and identifiability enables us to establish the identifiability in the linear case of the INAR equivalence class, which we develop in the next section.

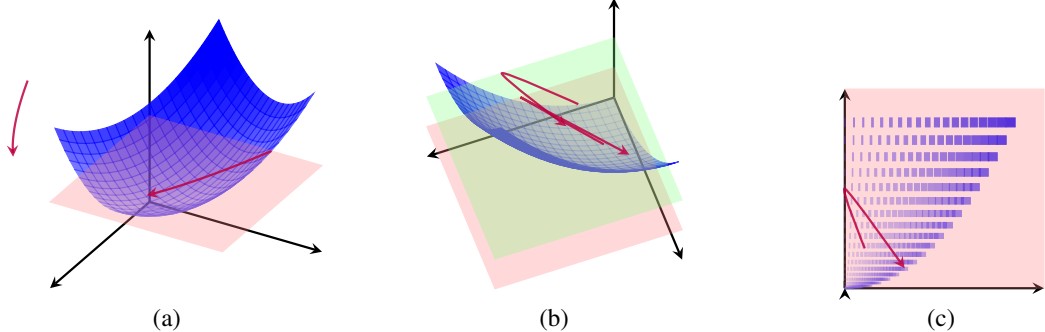

|       |       |       |
|-------|-------|-------|
| (a)   | (b)   | (c)   |

Figure 1: surfaces with generic hyperplanes (smooth gradient grid, 3D effect). (a) surface with one hyperplane;(b)Veronese with two hyperplanes;(c) surfaces with finite points

### 3.3 IDENTIFYING LINEAR MIXTURES

We begin with the identifiability result in the linear case. Specifically, we show that the full generative model $\Theta := (f, \Phi, U)$ can be recovered from a linear mixture $O_t = FZ_t^\Delta$, where $Z_t^\Delta$ denotes the weakly

equivalent class introduced in Section 3.1. For clarity, we drop the subsequence notation whenever the context is unambiguous. By Lemma 2, this latent process satisfies $\bar{Z}_t = \lambda_t + \epsilon$ with $\lambda_t = U + \Phi \star Z_t$.

In general, when causal variables are identified from the full distribution $P(O_t)$, sufficient variability of the latent distribution is required (Yao et al., 2022b; Song et al., 2023; Zhang et al., 2024). Unlike this setting, where only linear mixtures of $Z_t$ are observed, any variability in the latent distribution can arise solely from changes in the parameter $\Theta := (F, \Phi, U)$. This observation motivates the use of the algebraic structure of cumulants, which naturally captures distributional variability and provides a transparent interpretation of identifiability.

If the $d$-th order cumulant of $O_t$, denoted by $\kappa_d(O_t)$, also admits a unique decomposition, the reduced ideal $\mathcal{I}' := \langle K(\mathbb{I} - \Phi) - F \rangle$, where $K = F(\mathbb{I} - \Phi)^{-1}$, has degree at most one and admits a decomposition into a finite composition of prime ideals. The dimension of the solution space coincides with the dimension of the associated ideal $\mathcal{I}'$, thereby determining the identifiability of the full generative model. To this end, we need to control the geometry of the associated parameter space.

**Assumption 2.**

1. *$F$ is generic with the possible maximum rank almost surely.*

2. *There exists a set $D := \{d \in \mathbb{Z} \mid \kappa_{d+1}(O_t) = 0\}$ such that the collection $\{\kappa_d(O_t)\}_{d \in D}$ contains at least $p$ non-zero elements.*

3. *The ideal $\mathcal{I} := \langle F - K^{(k)}(\mathbb{I}_p - \Phi^{(k)}), \ k = 1, 2, \ldots, p \rangle$ is such that the space $\Theta$ is zero-dimensional.*

The three conditions are not independent of or parallel to each other. Instead, they are sequentially related, where each condition builds upon the preceding one, thereby establishing a stepwise progression toward full identifiability of the generative model. That is, each observed component $\kappa_d(O_t)$ has finite depth $d$ and the non-vanishing information up to this order is sufficiently rich to ensure identifiability via tensor decomposition: The proposed rank condition (1) is classic and results in a generically unique decomposition of each order $d$ tensor $\kappa_d(O_t)$, which uniquely recovers the component $K_j := (F(\mathbb{I} - \Phi)^{-1})_{t-s}^{(:,j)}$, for all $j \in [p]$ and all $s \in S := \{s : s < t\}$, up to permutation and rescaling. This holds for the Kruskal rank condition $\mathrm{krank}(V)$ (Kruskal, 1977; Lovitz & Petrov, 2023), which requires that each rank-1 component $(K_j)^{\otimes d}$ has no collinear columns in the ambient space[3](Wang & Seigal, 2024; Wang et al., 2025). Condition (2) guarantees that there are at least $p$ such points in the linear span of the outer space of $(K_j)^{\otimes d}$ that do not vanish, so that $F$ and $\Phi$ form a zero-dimensional parameter space. As a direct consequence of the combined results of condition (1), (2), and (3), we achieve full identifiability of the model.

**Theorem 1** (Linear identifiability of equivalent classes). *Under Assumption 1 and Assumption 2, the weakly-convergent equivalent class of the latent point process and the causal structure are identifiable up to component-wise scaling and permutation.*

We further remark that, once the generative model is identified, the causal variables can be sampled from the model $\Theta := (F, \Phi, U)$.

### 3.4 IDENTIFYING GENERIC NONLINEAR EQUIVALENT CLASSES

In this section, we relax the linearity assumption and demonstrate that the algebraic structure of the observed manifold can also ensure the identifiability under arbitrary nonlinear transformations (*cf.* the generic $F$ in Assumption 2). Specifically, we now assume that $f$ is generic (potentially injective), so that the mapping

---

[3]The *ambient space* is the higher-dimensional space in which a given variety or scheme is embedded, typically $\mathbb{A}^n$ or $\mathbb{P}^n$ in algebraic geometry. See details in Appendix C.1

$f : Z_t \mapsto O_t$ is well-defined and preserves the distinguishability of the latent representation. We formalize this in Assumption 3.

**Assumption 3.**

1. *Let $f$ be a generic $C^d$ map.*

2. *On the mixed cumulant manifold, the system admits a linear degeneration for which Assumption 2 holds for some order $d$ and $p$.*

The $C^d$ regularity assumption is strictly weaker than requiring $f$ to be a diffeomorphism, required by a spectrum of prior works, such as in (Song et al., 2023). Even in the presence of directional collapse within the latent space, the induced algebraic structure may still faithfully transmit the essential dependency relations to the observed domain. Unsurprisingly, one observes that the genericity of the mixing function $f$ is naturally satisfied when the elements of its Jacobian $J_f$ are sufficiently free functions (e.g., polynomials or $C^d$ functions). Indeed, the set of functions for which $J_f$ fails to be full rank corresponds to a proper algebraic subvariety of the function space, so that almost all choices of $f$ yield a full-rank Jacobian. Consequently, having functional (rather than constant) Jacobian entries increases the likelihood that $f$ is generic in the algebraic-geometric sense.

**Theorem 2** (Fully nonlinear identifiability of equivalent classes). *Under Assumption 1 and Assumption 3, the weakly-convergent equivalent class of the latent point process and the causal structure are identifiable up to component-wise transformation and permutation.*

**The intuition of our theorem.** The cumulant propagates the causal structure through nonlinear transformations, which enables the recovery of latent dependencies from partial algebro-geometric information on the mixed manifold $O_t$. We distinguish two cases: access to the full observational distribution $P(O_t)$, or access only to realizations drawn from a restricted distribution $P^R(O_t)$. In either case, the geometry of the cumulant can be checked: allowing tolerance of loss in distribution information up to a certain order, the cumulant admits a unique linear degeneration. This degeneration canonically determines a projective embedding of the Veronese variety, identical up to a component-wise scaling and permutation. Additionally, given multiple environments (interventions or variability) in condition (2), it follows that the full generative model cannot be contained in any hypersurface of positive dimension.

Importantly, the conditions stated in Assumption 2 are not merely technical assumptions, but collectively form a set of *necessary and sufficient conditions* for identifiability. Under the given data, there exists no alternative order $d_0 < d$ such that a strictly simpler cumulant manifold would still guarantee identifiability, unless additional data or interventions are introduced, as formalized below:

**Theorem 3.** *The identifiability result stated in Theorem 1, 2 holds if and only if the conditions in Assumption 2 are satisfied.*

## 4 MUTATE: ESTIMATING EQUIVALENT STOCHASTIC CAUSAL PROCESS

Building upon our identifiability theory, we formally introduce MUTATE (**MU**lti -**T**ime **A**daptive **T**ransition **E**ncoder), a novel Variational Auto-encoding framework for estimation of latent multivariate stochastic point processes. Importantly, the framework is modular and can be readily adapted to other types of stochastic processes with suitable modifications. We highlight two core features of MUTATE, each addressing a key challenge related to identifiability. First, the central objective of our method is to recover the latent realization sequence $\{Z_{t_0}^\Delta, Z_{t_1}^\Delta, \ldots, Z_T^\Delta\}$ from multiple unstructured observational sources $\{O_t\}_{t=0}^T$. Unlike prior frameworks that rely primarily on time-stamp conditional independence to enforce latent structure, our approach accounts for the nature of progressively adaptive stochastic processes.

In such systems, the filtration $\mathcal{F}_T$, which captures the intrinsic history of the process, is defined as $\sigma\left(\bigcup_{0<t<T}\sigma(Z_t^\Delta)\right)$ and grows strictly over time. As shown in Figure 2, this dynamically expanding information structure poses unique challenges for both identifiability and representation learning, which MUTATE is explicitly designed to address. In addition, to leverage mutually independent noise, we employ a power spectral density (PSD) decomposition module in the joint optimization of parameters, which automatically enforces global whiteness of the noise.

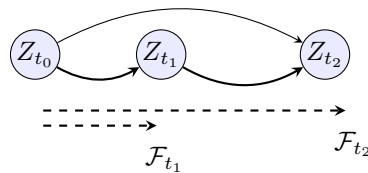

Figure 2: Visualization of information loss in increasing filtration.

### 4.1 Time adaptive transition module

To infer this latent structure from observed data, we first employ an encoder $q_\phi(Z_t^\Delta|O_t^\Delta)$ to learn the estimated latents $Z_t^{(\Delta)} \sim q_\phi(Z_t^\Delta \mid O_t^\Delta)$ as commonly applied in representation learning frameworks. Recall that the latent process is modeled as $Z_t = \Phi \star Z_t + R_t$, where $\Phi$ denotes a global convolution kernel and $R_t = U + \epsilon_t$ is a residual process. Under our weak convergence condition, $Z_t$ receives a well-structured representation $Z_t = (\mathbb{I} - \Phi)^{-1} \star (U + \epsilon_t)$, which is also a $(U + \epsilon_t)$-measurable process with tractable Power Spectrum Density (PSD)

$$S_{Z_t^{(\Delta)}}(w) = (I - \Phi)^{-1}\Sigma(I - \Phi)^{-H} \tag{6}$$

where the baseline $U$ is treated as a learnable parameter in the model and $A^H$ is the Hamilton conjugate transpose of $A$. $w$ is a continuous frequency variable. The inverse mapping $f_O^{-1}$ is an encoder with trainable parameters of neural networks. Importantly, the learned functional $f$ maps the observation $Z_t$ to a space of independent varying noise through the designated PSD decomposition that enforces $\Sigma$ being a diagonal matrix and recursively infers $H^\dagger = (\mathbb{I} - \Phi)^{-1}$. Then, the evaluated prior from the PSD module is sent to calculate the KL divergence. Decomposing $S(w)$ in Eq. (6), each component of transitions satisfies $\log p(Z_t^{(\Delta)}|\mathcal{F}_{t-}) = \log p[(I - \Phi) \star R_t^\Delta]$, which is the main part of the latent prior estimation. Our model is trained based on the Variational Auto-encoding framework (the detailed derivation is referred to Appendix E). Therefore, we aim to maximize the log likelihood of observation $\log p_{data}(X)$ through the evidence lower bound (ELBO):

$$ELBO = -\mathcal{L}_{recon} - \alpha\mathcal{L}_{KL}$$

$$= \mathbb{E}_{z\sim q(Z_t|O_t)}\left[\log p(O_t|Z_t) - \log q(Z_t|O_t)\right] + \mathbb{E}_{z\sim q(Z_t|O_t)}\left[\sum_{\mathcal{F}_0^+}^{\mathcal{F}_T}\log p(Z_t^{(\Delta)}|\mathcal{F}_{t-})\right]$$

$$+ \mathbb{E}_{z\sim q(Z_t|O_t)}\left\{\sum_{\mathcal{F}_0^+,Z_t,N\in(N_0,T)}^{\mathcal{F}_T}\log p\left[\mathcal{N}(\hat{U}\text{PSD}_{Z_t}(H(0)), \frac{1}{N}\sum_{k=0}^{N-1}S_{Z_t}(w_k))\right]\right\} \tag{7}$$

## 5 Experimental results

We simulate multivariate point processes and their converging equivalent class $\mathcal{Z}_t$ extensively studied in our identifiability theory. We sample all point processes using the Poisson Superposition method (rejection sampling from the upper bound of conditional intensity (Cinlar & Agnew, 1968; Albin, 1982)) to mimic highly dynamic changes in conditional intensity, and capture denser information contained in stochastic processes. Then we create corresponding converging classes as a proof-of-concept validation: A total of 20,000 latent trajectories are sampled for each of the five kernel functions—exponential, power-law, rectangular, simple nonlinear, and flexible mixing—under two noise regimes: heterogeneous noise and Gaussian mixture noise.

To illustrate the latent events underlying the unstructured data, we also simulate stochastic dynamics for biological data using SERGIO (Dibaeinia & Sinha, 2020), a GRN-guided gene expression simulator used in Lorch et al.'s Lorch et al. (2024) causal modeling as well. All observations $O_t$ is obtained from latents $Z_t$ through MLP and LeakyReLU nonlinearity mixing. A detailed simulation procedure is included in E.1.

To validate our identifiability results, we evaluate against several representative baselines, including TDRL (Yao et al., 2022a), BetaVAE (Higgins et al., 2017), SlowVAE (Klindt et al., 2021), and PCL (Hyvarinen & Morioka, 2017). Among them, PCL and TDRL incorporate temporal dependencies by leveraging historical information and explicitly enforcing conditional independence among latent variables to recover underlying dynamics. In contrast, BetaVAE and SlowVAE assume independent latent components and disregard any time-delayed mechanisms.

**Evaluation metrics**   We validate our method on both synthetic and real-world datasets. For selected baseline models, we adapt the factorized inference module—commonly employed in nonlinear ICA—to support a deeper composition of intrinsic filtration, enabling the modeling of complex real-world dynamics. On synthetic datasets, we assess identifiability using the Mean Correlation Coefficient (MCC), a standard metric that quantifies the recovery accuracy of latent variables. Specifically, MCC is computed by averaging the absolute correlations between the ground-truth and inferred latent components after solving a linear assignment problem to handle permutation indeterminacy.

**Results**   Performance of all baselines and our model is shown in Table 1 with extended results reported in Table A2. During training, both BetaVAE and SlowVAE tend to converge prematurely, typically reaching a local optimum within the first epoch and triggering early stopping. This behavior highlights their limitations in modeling temporal structures essential for identifying latent event-driven processes. TDRL performs reasonably when the lag module is set to a longer one (we use $L = 9$ in experiments) since it can harness shorter temporary contextual information. It is noticed that our identifiability can be readily applied to the prior framework by either adding the domain index in synthetic datasets or modulating the distribution shifts that change pairs of edges in the latent space. However, we also realize that the fully non-parametric setting is hard to interpret since our identifiability avoids this.

Table 1: MCC Scores with standard deviations for five kernels

| Method | Ave.($\uparrow$ better) | Exponential | Powerlaw | Rectangular | nonlinear | nonparametric |
|---|---|---|---|---|---|---|
| TDRL | 0.599 | 0.593±0.028 | 0.609±0.043 | 0.618±0.056 | 0.556±0.016 | **0.616**±0.043 |
| BetaVAE | 0.141 | 0.153±0.863 | 0.128± 0.077 | 0.128±0.078 | 0.146±0.108 | 0.149±0.096 |
| SlowVAE | 0.115 | 0.108±0.075 | 0.104±0.073 | 0.104±0.073 | 0.126±0.074 | 0.131±0.076 |
| PCL | 0.375 | 0.395±0.034 | 0.330±0.029 | 0.330±0.029 | 0.414±0.028 | 0.404±0.028 |
| **MUTATE**(ours) | **0.837** | **0.853**±0.218 | **0.938**±0.036 | **0.879**±0.102 | **0.921**±0.029 | 0.598±0.013 |

## 6   CONCLUDING REMARK

This work extends causal representation learning framework to stochastic causal dynamics (i.e., multivariate Hawkes Processes), a topic not yet covered in current CRL literature. We show that, under sufficiently generic conditions, the generative model of a latent stochastic point process can be fully identified. Our results bridge the gap between stochastic modeling and causal representation. We also propose a novel framework to estimate the latent point processes. However, our work avoids the worst, the most complex scenario for a fully nonparametric kernel, which, in empirical practice, can be replaced with a simpler kernel. Future directions may include solving this condition and causal representation learning for stochastic differential processes that manifest in rich scientific questions.

## ETHICS STATEMENT

This paper does not arouse any significant concerns of ethics, such as human subjects, harmful experiments, or violation of social fairness.

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

*Supplement to*

# "Causal Representation Meets Stochastic Modeling"

## A    USEFUL LEMMATA

### A.1    PRELIMINARY LEMMAS

The identifiability stated in Theorem 2 builds upon Lemma 1 and Lemma 2 that discuss a wide range of convergence conditions under a space metric and a topological space. For seamless understanding, we introduce those basic but crucial concepts with illustrations.

**Lemma A.1** (Weak Convergence Billingsley (1999)). *Let $(S, \mathcal{S})$ be a Polish space equipped with its Borel $\sigma$-algebra, and let $\{Z_n\}_{n \in \mathbb{N}}$ and $Z$ be $S$-valued random elements defined on a common probability space. Then the sequence $\{Z_n\}$ converges in distribution (i.e., weakly) to $Z$, denoted $Z_n \Rightarrow Z$, if and only if*

$$\lim_{n \to \infty} \mathbb{E}[f(Z_n)] = \mathbb{E}[f(Z)]$$

*for all bounded continuous functions $f : S \to \mathbb{R}$.*

The proof and demonstration of this lemma is classic in basic probability that we omit here. The weak convergence, in most cases, corresponds to the convergence of finite dimension distribution of a process or a variable.

**Lemma A.2** (Tightness of the Measure $\mathbb{P}_{Z^{(\Delta)}}$). *Let $\{Z_n^{(\Delta)}\}_{n \in \mathbb{N}}$ be a sequence of $S$-valued random elements (e.g., stochastic processes or path evaluations) indexed by $\Delta$ and defined on a Polish space $S$ with Borel $\sigma$-algebra. Then the sequence of corresponding probability measures $\{\mathbb{P}_{Z_n^{(\Delta)}}\}$ is tight. In particular, any subsequence admits a further weakly convergent subsequence.*

Tightness of a sequence of probability measures ensures the existence of well-behaved subsequences: every subsequence admits a further weakly convergent subsequence. This property is particularly useful in Polish spaces, where tightness is equivalent to relative compactness (precompactness) under the weak topology. However, it is important to note that precompactness does not imply full compactness; in general, a tight sequence need not converge without an additional uniqueness or limit identification argument. Thus, tightness provides necessary control over subsequential behavior, but does not guarantee full convergence of the entire sequence.

**Lemma A.3** (Higher-Order Moment Bound Implies Lower-Order Bounds). *Let $\{Z_n\}_{n \in \mathbb{N}}$ be a sequence of real-valued random variables defined on a common probability space. Fix an integer $d > 0$. Suppose there exists a constant $C > 0$ such that*

$$\sup_{n \in \mathbb{N}} \mathbb{E}[|Z_n|^d] \leq C.$$

*Then for any $0 < p < d$, there exists a constant $C_p > 0$ such that*

$$\sup_{n \in \mathbb{N}} \mathbb{E}[|Z_n|^p] \leq C_p.$$

### A.2    DISCUSSION OF POINT PROCESS

To motivate, we recall the dynamics of a stochastic point process, a family of non-homogeneous point processes with unfixed intensity, by the following example:

**Motivating example.**    We start with dynamics that can be seen in numerous scientific fields. Consider a system with two latent provably existing disease-associated processes (i.e., biological mutations) $M = (M_t^1, M_t^2)$ for $0 \leq t < \infty$. By nature of point processes, we denote by $\phi$ a kernel function that conveys impacts to different processes and $\lambda_t$ the conditional intensity of the process at time $t$. Therefore, the

conditional intensity vector for each type of mutation is expressed as

$$[\lambda_1(t), \lambda_2(t)] = \left[ u_1 + \int_0^t \sum_{j=1}^2 \phi_{1 \leftarrow j}(t - t') \, dM_{t'}^j, u_2 + \int_0^t \sum_{j=1}^2 \phi_{2 \leftarrow j}(t - t') \, dM_{t'}^j \right]$$

The interpretation is natural: for one type of mutation $M_t^1$ (potentially on different genes), all past existing mutants before $t$ will be contributing to the higher probability (described as intensity $\lambda_t^1$) of future mutations through the multiplying effect with a *time-invariant* kernel. Since kernel function relays possible impacts to all participating processes, causal influences are fully captured in kernel matrices along with their evolutions. $\phi$ denotes one element of $\Phi_t$ and $M_t^i(\Delta)$ is the measure of a counting process $M_t^j$, also the integral measure in Itô calculus, distinct from deterministic calculus. Note that one *cannot* perform the standard calculus operation on this form since the integral element $M_t^j(\Delta)$ is not differentiable as it is in the Riemann–Stieltjes integral at any point, since it has stochastic jumps at each time step. The counting process $M_t^i$ and the conditional intensity $\lambda_t^i$ satisfy: $M_{t+\Delta t}^i - M_t^i = M_{\Delta t}^i = M_t^i(\Delta)$ and $\lambda_t^i = \frac{\mathbb{E}[M_t^i(\Delta)|\mathcal{F}_t]}{\Delta t}$. Dividing time into sufficiently small intervals $[t_k, t_{k+1})$ gets us the form of discrete intensity:

$$\lambda_k^i = u^i + \sum_{j=1}^p \sum_{s < k} \phi_{i \leftarrow j}(k - s) \, \Delta M_s^i \tag{A.1}$$

We highlight the difference between the kernel matrix $\begin{bmatrix} \phi_{1 \leftarrow 1}(t') & \phi_{1 \leftarrow 2}(t') \\ \phi_{2 \leftarrow 1}(t') & \phi_{2 \leftarrow 2}(t') \end{bmatrix}$ and the regular coefficient matrix $A_k$ for step $k$: the kernel matrix encodes not only the model information but the evolution of the system. For example, given $t_1, t_2$, $(i, j)$ entry of the kernel matrix varies due to the generative nature of $\phi$. $(i, j)$ entry of $A_k$, however, just captures the effect of the variable $i$ on $j$ regardless of the evolution direction in the system.

**Assumption A.1** (Stability and stationary Increment, Proposition 1 in (Bacry et al., 2015)). *The process $N_t$ has asymptotically stationary increments, and intensity $\lambda_t$ is asymptotically stationary if the kernel satisfies the assumption:*

$$\rho_{\Phi(t)} = \|\Phi(t)\| = \int_0^t |\Phi(t)| \, dt \text{ has spectral radium smaller than 1} \tag{A.2}$$

Assumption A.1 gives a necessary condition so that the point process has stable, stationary increments in its intensity. In particular, it means the entire process tends to be stable with an unknown but fixed expectation of the conditional intensity $\mathbb{E}[\lambda_t^i] = \Lambda^i$. Restricted by the stationary increment assumption, the existence of the corresponding process is ensured by Lemma 3. To illustrate those conditions, we show a simpler version kernel in Example 1.

*Example* 1. Consider a point process whose kernel functions relay causal influence with an exponential decay to other processes. The generating process thus be accordingly

$$\lambda_t^i = u^i + \sum_{j=1}^p \int_0^t \alpha^{ij} e^{-\beta(t-t')} \, dN_{t'}^j$$

shows the exponential kernel triggers influences that are sustaining but decaying as time proceeds. Technically, the induced causal influences, although decaying from inside the system dynamics, will not disappear unless the causal strength $\alpha = 0$ for all $j$.

**Lemma 3** (Proposition 6 in Kirchner (2016)). *If all conditions and results in Assumption A.1 hold almost everywhere, there exists only one determined process whose dynamics match observations with regard to $\Lambda^i$.*

**Convoluted Kernels and Intensity.** We evaluate stochastic integrals in continuous time, where kernel-induced causal influences decay smoothly over time. Suppose $t < t'$ denotes any time before $t$. Then, the kernel vector evaluated at $t'$ for $p$ stochastic processes is given by:

$$\phi_i(t - t') = (\phi_{i,1}(t - t'), \phi_{i,2}(t - t'), \phi_{i,3}(t - t'), \cdots, \phi_{i,p}(t - t'))$$

This leads to an integral in the form of

$$I_i(t) = \int_0^t \sum_{j=1}^2 \phi_{ij} \, dN_t^j = \int_0^t \sum_{j=1}^2 \phi_{ij}(t - t') \, d \begin{bmatrix} N_1(t') \\ N_2(t') \\ \vdots \\ N_p(t') \end{bmatrix} = \int_0^t \phi_i(t - t') \, dN_{t'} \tag{A.3}$$

Collecting $p$ integrals

$$\begin{bmatrix} I_1(t) \\ I_2(t) \\ \vdots \\ I_p(t) \end{bmatrix} = \int_0^t \begin{bmatrix} \phi_{11}(t - t') & \cdots \\ \phi_{21}(t - t') & \cdots \\ \vdots & \ddots \\ \phi_{p1}(t - t') & \phi_{pp}(t - t') \end{bmatrix} d \begin{bmatrix} N_1(t') \\ N_2(t') \\ \vdots \\ N_p(t') \end{bmatrix} = \int_0^t \Phi(t - t') dN_{t'} = \int_0^t \Phi(t') dN_{t - t'}$$

which concludes it as the convolution of the kernel matrix $\Phi$ and the stochastic jump vector $dN$ up to time $t$

$$I(t) = \Phi_t \star dN_t \tag{A.4}$$

### A.2.1 REMARKS ON THE FILTRATION

In probability theory, the filtration $\mathcal{F}_t$ is defined as the smallest $\sigma$-algebra that renders the intensity process $\lambda_t$ to be $\mathcal{F}_t$-adapted and measurable. This filtration is constructed by the minimal closure under set operations (e.g., union, intersection) over past events, ensuring that $\lambda_t$ evolves consistently with the observable history (Hawkes, 1971; Daley & Vere-Jones, 2005). Therefore, for any filtration as its internal history, we have $\mathcal{F}_s \subseteq \mathcal{F}_t$, for $s \leq t$. Note that the filtration $\mathcal{F}_t$ may theoretically differ from the intrinsic history $\mathcal{H}_t$, which introduces additional challenges in the evaluation and modeling of point processes. For a comprehensive discussion on scenarios where $\mathcal{F}_t$ and $\mathcal{H}_t$ are defined differently, we refer the interested reader to (Daley & Vere-Jones, 2005). We occasionally overload the notation $dN_t^i$, which represents an integral element in stochastic calculus, to distinguish it from its deterministic counterpart. Despite potential similarities in notation, they are fundamentally different: while standard calculus considers infinitesimal increments over fixed mesh widths (e.g., $dg(x)$ as $\Delta t \to 0$), the increment $dN_t^i$ is a random variable governed by the stochastic process. Specifically, its realization at each infinitesimal interval is drawn from a Bernoulli process with intensity $\lambda_t^i$, such that $\mathbb{P}(dN_t^i > 0 \mid \mathcal{F}_t) = \lambda_t^i, dt$. In contrast to deterministic differentials, $dN_t^i$ encapsulates the uncertainty of event occurrences within each interval. The kernel matrix $\Phi_t$ consists of time-decaying kernel functions that transmit the influence of past events across processes. It captures both time-delayed and causal dependencies, and plays a central role in modeling self-exciting or mutually-exciting dynamics.

### A.3 CUMULANTS AND TENSORS

**Cumulant tensor notation.** The $d$-th order cumulant tensor of a random vector $X \in \mathbb{R}^p$ is denoted $\kappa_d(X) \in \mathbb{R}^{p \times \cdots \times p}$, and is symmetric in all modes. In ICA and CRL settings, cumulants of independent components often admit a CP form:

$$\kappa_d(X) = \sum_{r=1}^R \lambda_r \cdot v_r^{\otimes d},$$

where $v_r \in \mathbb{R}^p$ and $\lambda_r \in \mathbb{R}$. This structure enables identifiability of latent sources from cumulant information.

**Tensor notation and operations.** We denote an order-$d$ tensor as $\mathcal{T} \in \mathbb{R}^{I_1 \times I_2 \times \cdots \times I_d}$. The outer product $u^{(1)} \otimes \cdots \otimes u^{(d)} \in \mathbb{R}^{I_1 \times \cdots \times I_d}$ produces a rank-1 tensor with entries:

$$\mathcal{T}_{i_1,\ldots,i_d} = u_{i_1}^{(1)} \cdots u_{i_d}^{(d)}.$$

Given a tensor $\mathcal{T} \in \mathbb{R}^{I_1 \times \cdots \times I_N}$ and a matrix $U \in \mathbb{R}^{J \times I_n}$, the *mode-n* product $\mathcal{T} \times_n U \in \mathbb{R}^{I_1 \times \cdots \times I_{n-1} \times J \times I_{n+1} \times \cdots \times I_N}$ is defined as:

$$(\mathcal{T} \times_n U)_{i_1,\ldots,i_{n-1},j,i_{n+1},\ldots,i_N} = \sum_{i_n=1}^{I_n} \mathcal{T}_{i_1,\ldots,i_N} \cdot U_{j,i_n}.$$

## B  PROOF OF SUPPORTIVE RESULTS

A point process is associated with a counting process $N_t$, which arises from its random measure over a measurable space $S$. Taking the limit as the mesh width tends to zero yields the conditional intensity process:

$$\lambda_t = u + \Phi_t \star dN_t. \tag{B.1}$$

Even a univariate point process does not satisfy the autoregressive property, as the intensity $\lambda_t$ is itself stochastically driven by internal dynamics. This makes the process self-exciting and adapted to the filtration $\mathcal{F}_t$. Such a non-autoregressive structure poses challenges in formulating a causal model for a collection of stochastic intensity variables $\{\lambda_t^i\}_{t \in T, i \in [p]}$. To model the relationship between the stochastic jumps $dN_t$ and their conditional intensities $\lambda_t$, we aim to find a representation of the form $dN_t = f(\lambda_t)$ that is compatible with a latent causal model, which is what our weak convergence class aims at.

**Remark B.1.** *The relationship between $dN_t$ and $\lambda_t$ can be written more compactly. By definition, the conditional intensity satisfies:*

$$\lambda_t = \frac{\mathbb{E}[N_{t+dt} - N_t \mid \mathcal{F}_t]}{dt} = \frac{\mathbb{E}[dN_t \mid \mathcal{F}_t]}{dt}.$$

*This naturally leads to a decomposition:*

$$dN_t = \lambda_t dt + dM_t,$$

*where $dM_t$ is a local martingale capturing the stochastic deviation from the conditional expectation. This decomposition is analogous to the standard regression form $Y = \mathbb{E}[Y \mid X] + \epsilon$, with the filtration $\mathcal{F}_t$ taking the role of covariates and $dM_t$ representing a stochastic error term.*

Under the assumption of a causally sufficient system, the residual noise vector $R_t = (r_t^1, r_t^2, \ldots, r_t^p)$ is component-wise independent. This specific representation of the point process facilitates further analysis using operator-theoretic tools. The conditional intensity $\lambda_t$ can be interpreted as a short-term estimate of the expected number of events in process $i$ at time $t$. This idea is formalized in Fact 1, whose proof is provided in Appendix B.

**Fact 1.** Let $N_t$ be a counting process with conditional intensity defined in Eq. (1). Then the residual between the discrete-time process $N_t^\Delta$ and the approximation $\lambda_t dt + dM_t$ is uniformly bounded with high probability. Specifically, with probability at least $1 - \epsilon$, the following holds:

$$\left| N_t^\Delta - \lambda_t dt - dM_t \right| = o\left( \log\left( \frac{f(\Lambda)}{T^2} \right) \right).$$

### B.1 PROOF OF LEMMA 1

This lemma significantly constitutes the reasoning chains that lead to our identifiability results. We restate the original statement to provide more details and background on the point process and theory of weak topology and convergence.

**Lemma B.1** (Bounding Point Process in intensity, *constructive*). *For a measurable mapping $N^\Delta : (\Omega, \mathcal{F}) \to (M_p, \mathcal{M})$ such that $\omega \mapsto N(\omega)$ is a point process at scale $\Delta$. Let $\Delta$ be the control operator for any subsequence of its point process. Consider $A \in \mathcal{B}$ generated by the topology $\mathcal{M}_p := \mathcal{B}(M_p)$. $\lambda$ and $\epsilon$ is defined on this metric space. If $\lambda$ satisfies the stationary increment condition, then we can establish the weak convergence of the constructed equivalent class:*

$$\sum_{k:k\Delta \in A} \lambda_k^{(\Delta)} + \epsilon_k^{(\Delta)} \overset{w}{\Rightarrow} N(A) \ \ for \ \lambda_k^\Delta = \lim_{\Delta \to 0} \lim_{\delta \to 0} \frac{\mathbb{E}[dN^\Delta|\mathcal{F}]}{\delta}$$

*Proof.* We organize our proof into three steps. First, a trivial case can be readily justified when $\Delta = 1$, consistent with the discrete time autoregressive model. Second, to keep the conditional intensity well-behaved, the numerator and denominator should have simultaneous and proportional changes. By constructing a sub-sequence $N_t^{(\Delta)}$, where $t \in \mathcal{B}(M_p)$ is a $\sigma$- algebra generated by a weak topology $\mathcal{M} : \mathcal{B}(M_p)$, we show that the trivial convergence in step 1 can be extended to the case of $\Delta \in (0, 1)$. Last, a weak convergence to a continuous autoregressive causal process is established through the property of a uniformly tight measure.

We start the proof with a trivial case. If $\delta = \Delta_1 = 1$, the conditions always trivially hold. In this case, we only need to show $N_t^{(\Delta_1)} = \lambda_t^{(\Delta_1)} + R_t$ by simply using the tower rule. Therefore, our proof gives more attention to the non-trivial case for $\delta \neq 1$.

*Case 2: $\delta \in (0, \Delta_1)$*

The reasoning of this case becomes more complicated if the time step operator used for generating sub-sequences proportionally shrinks to a sufficiently small unit $(0, \Delta_1)$. We rewrite the approximating sequence $N$ to leverage the metricizability of the space. Since we work in a Polish space, the Borel $\delta$-algebra is countably generated and the space is separable and metrizable. Given a measurable set $A \in \mathcal{B}$, and a metric $\rho$, define the open $\delta$-neighborhood as:

$$\mathcal{A} = A^\delta := \{x \in \mathbb{R}^d : \rho(x, A) < \delta\}$$

By outer regularity of Borel probability measures on Polish spaces, for every $\epsilon > 0$, there exists a countable collection of open sets $\{A_i\}_{i \in \mathbb{N}}$ such that $\bigcup_i A_i \supset \mathcal{A}$ and $\sum_i \mu(A_i \setminus A) < \epsilon$. This allows us to approximate any compact subset from outside using open sets with arbitrarily small excess mass and ensures the approximating sequence is defined on a non-decreasing base. We paraphrase the convergence as

$$\sum_{k:k\Delta \in A} \lim_{i \to \infty} \frac{\mathbb{E}[N^\Delta(A_i)|\mathcal{F}]}{|A_i|} + \epsilon_k^{(\Delta)} \overset{w}{\Rightarrow} N(A) \ \ \text{for } \Delta \to 0 \tag{B.2}$$

The equation above is adapted from the continuous-time intensity for point processes. However, it requires us to work with two limit conditions for $A_i$ with the $1/k$ closed ball shrinking to zero measure and for the subsequence operator $\Delta$ approaching 0. A common method is to ensure dominated and uniform convergence of the limit. To harness information regarding the intensity in our convergence to a more generalized process, we first work with only the operator $\Delta$ to induce the same time scale of intensity function. Therefore, we have the equivalent condition

$$\sum_{k:k\Delta \in A} \frac{\mathbb{E}[\sum_{k=1} Z_k^\Delta - \sum_{k=1} Z_{k-1}^\Delta |\mathcal{F}]}{|A_i|} + \epsilon_k^{(\Delta)} = \sum_{k:k\Delta \in A} \frac{\mathbb{E}[Z^\Delta(\Delta)|\mathcal{F}]}{\Delta} + \epsilon_k^{(\Delta)} \overset{w}{\Rightarrow} N(A) \ \ \text{for } \Delta \to 0 \tag{B.3}$$

We remove the limit condition as it is clear that $|A_i|$ is of measure zero when $\Delta = 0$, which ensures the alignment between our topological property and plausibility to analyze only subsequences in the sequel. According to Lemma 2 of (Kirchner, 2016), for any compact interval $[a,b]^{(\delta)}$ with the number of bins $[b-a]/\delta$, $\mathbb{E}[N^{(\delta)}([a,b])] < (b-a+2)(I - G^{(\delta)}(a,b))^{-1}\Lambda$ where $G(a,b) = \int_a^b \Phi(s)ds$ is a solution of the stochastic differential equation systems

$$\mathbb{E}[\lambda([a,b])] = \mathbb{E}[u + G(a,b)\Lambda], \text{ for } \mathbb{E}[\lambda(a,b)] = \Lambda$$

Note that, by reapplying the tower rule, Eq. (B.2) implies:

$$\lim_{\delta \to 0} \mathbb{E}[N_A^{(\delta \in (0,1))}] \to \lim_{\delta \to 0} \mathbb{E}\big[\frac{\mathbb{E}[N_A^{(\delta)}|\mathcal{F}_t]}{\delta}\big]$$

Next, we show the necessity of tightness of the corresponding probability measure $\mathbb{P}^\Delta$ for the left-hand of Eq. (B.3) to achieve the desired convergence. Without loss of generality, we consider a nonparametric intensity function $\lambda_t = \psi(u + \int \phi(t-s)Z^\Delta(s)\, ds)$. Consequently, $\mathbb{E}[\lambda_t] = \Lambda$ and $\mathbb{E}[\lambda_t] = \mathbb{E}[\psi(u + \int \phi(t-s)Z^\Delta(s)\, ds)]$. Assume that $\psi$ is $\alpha$-Lipschitz and $\alpha\|\phi\|_1 < 1$ (Brémaud & Massoulié, 1996), so the mapping $F(\Lambda) = \psi(u + \|\phi\|_1\Lambda)$ is a contraction on $\mathbb{R}_+$. By Banach's fixed-point theorem, there exists a unique solution $\Lambda^\star$ to the equation:

$$\Lambda^\star = \psi(u + \|\phi\|_1\Lambda^\star)$$

Formally, this can be rearranged as:

$$\psi^{-1}(\Lambda^\star) - \|\phi\|_1\Lambda^\star = u \quad \implies \quad \Lambda^\star = \big(\mathrm{id} - \|\phi\|_1 \cdot \psi^{-1}\big)^{-1}(-u)$$

provided that $\mathrm{id} - \|\phi\|_1 \cdot \psi^{-1}$ is invertible on the image of $\psi$.

To control the tail probability, we apply Markov's inequality:

$$\mathbb{P}\left(\sum_{k:k\Delta \in A} \frac{\mathbb{E}[Z^\Delta(\Delta)|\mathcal{F}]}{\Delta} + \epsilon_k^{(\Delta)} > M_\varepsilon\right) \le \frac{\mathbb{E}[\sum_k \Lambda^\Delta]}{M_\varepsilon} \le \frac{(b-a+2\delta) \cdot \Lambda^\star}{M_\varepsilon}$$

Here, we define:

$$M_\varepsilon := \frac{(b-a+2\delta) \cdot \Lambda^\star}{\varepsilon} \quad \text{where } \Lambda^\star = \psi(u + \|\phi\|_1\Lambda^\star)$$

This choice ensures the upper bound remains within the prescribed $\varepsilon$-level for all $\Delta \in (0, \Delta_1)$. Since the only thing we need is the precompactness, we will not establish any tighter bound. Tightness of measure, as presented in Lemma A.2, indicates we can always find a subsequence $\lambda_{k_n}^\Delta + \epsilon_{k_n}^\Delta$ in $\lambda_k^\Delta + \epsilon_k^\Delta$ converges weakly to a sequence $\lambda^\star + \epsilon^\star$. This weak convergence of subsequences, however, cannot control the limit uniqueness for each sequence. Therefore, we also should further control the limiting behavior of each sequence by uniform convergence of the characteristic functional defined by the approximating process and the target process, which corresponds to the central idea of Lemma 2. $\qquad\square$

## B.2 PROOF OF LEMMA 2

**Lemma B.2** (Converging to the equivalent class, constructive for finite dimension distribution). *Under Lemma 1 and its constructive version, each subsequence $N_k^\Delta(A_i)$ defined on the measure $\mathbb{P}^\Delta$ converges weakly to a limit process $N_t$, and this limit exists and is unique.*

*Proof.* This proof is tedious but straightforward, adapted from the uniform convergence of the finite-dimensional moment generating function (MGF) for any compactly supported continuous function $f$. The procedure is organized as follows: the sub-sequence convergence in Lemma 1 chooses an arbitrary sequence with the characteristic function $\Psi(N_k^\Delta)$ that also converges to $\Psi(N^{\Delta \to 0})$. Provided the process has a bounded second variation, ensured by Lemma A.3, the subsequence has the same limit as the original process $N$. Equivalence between characteristic functions indicates uniform convergence in their behavior. $\qquad\square$

## C    PROOF OF IDENTIFIABILITY THEORY

### C.1    NOTATIONS

**Projective space of causal representation**    For the rest of this proof, we study the algebraic structure of the proposed latent causal models. We work in the complex projective space $\mathbb{P}^n$, also written as $\mathbb{CP}^n$, which formalizes the usual identifiability convention that matrices are considered equivalent up to a nonzero scalar multiple. Concretely, for a vector $v \in \mathbb{C}^{n+1} \setminus \{0\}$, its equivalence class in $\mathbb{P}^n$ is $[v] = \{\lambda v \mid \lambda \in \mathbb{C} \setminus \{0\}\}$. For example, given an algebraic object $V := V(f(x,y)) \subseteq \mathbb{P}^1$, where $V$ is the vanishing locus of a *homogeneous* polynomial $f(x,y)$, each point $[x : y] \in \mathbb{P}^1$ corresponds to a line through the origin in $\mathbb{C}^2$. Under the usual identification $\mathbb{P}^1(\mathbb{C}) \simeq \hat{\mathbb{C}}$ (the Riemann sphere), each line intersects the unit sphere $S^2 \subset \mathbb{R}^3$ in two antipodal points. Therefore, topologically, we have $\mathbb{P}^1 \cong S^2$. Unless noted otherwise, all rings considered in this paper are assumed to be commutative, Noetherian (finitely generated), and to possess a multiplicative identity. In particular, we focus on rings such as $K(x,y,z)$ and their subrings, e.g., $f(x,y) \subseteq K(x,y,z)$, which are always understood to satisfy these properties. Additional assumptions, such as being an integral domain or a field, will be explicitly stated when required.

**Genericity and Full Rank.**    Throughout this work, we regard matrices

$$F \in \mathbb{C}^{n \times p} \quad \text{as points} \quad P = (p_0, p_1, \ldots, p_{np-1}) \in \mathbb{P}^{np-1} \text{ or } \mathbb{CP}^{np-1},$$

where the entries of $F$ are identified with the homogeneous coordinates of $P$. A point $P$ is said to be *generic* if there exists no non-zero polynomial $f \in K[x_0, \ldots, x_{mn-1}]$ such that $f(p_0, \ldots, p_{mn-1}) = 0$. Equivalently, the coordinates of a generic point are algebraically independent over the base field $K$. Genericity implies that the corresponding matrix $F$ is of full rank almost surely, since the vanishing of any minor corresponds to the zero locus of a non-zero polynomial, which a generic point cannot lie on. However, the converse is generally false: a matrix can be of full rank without its entries being algebraically independent. Therefore, the set of generic points encompasses a broader range of matrices than merely the injective or invertible ones.

**Low-dimension embedding**    To embed algebraic objects arising from our models, we introduce a higher-dimensional projective space $\mathbb{P}^N$ with $N \geq n$, called the ambient space, into which $\mathbb{P}^n$ is naturally included (Hartshorne, 1977, Ch. I). We say that a projective variety $V \subset \mathbb{P}^n \subset \mathbb{P}^N$ has codimension $N - n$ in $\mathbb{P}^N$, meaning that $\dim \mathbb{P}^N - \dim V = N - n$, where $\dim V$ denotes the projective dimension of $V$. Under these conventions, all algebraic objects associated with the latent causal models are understood projectively, so that equivalence under scaling is built into the framework.

### C.2    PROOF OF THEOREM 1

We decompose the proof of Theorem 1 by walking through the algebro-geometric viewpoints that are increasingly related to the full identification of the entire generative model. First, we obtain algebraic cumulants for infinite-order INAR models via a multi-linear transformation, which guarantees the recovery of $K = F(\mathbb{I}_p - \Phi)^{-1}$, where the factor $[\![K^{(1)}, K^{(2)}, \ldots, K^{(p)}]\!]$ lies on a Veronese variety, provided Lemma C.1 holds. It follows that a topologically ordered representation $\mathscr{G}$ is provided by embedding an arbitrary $\Phi(\tau)$ to a larger ambient space $\mathbb{R}^{2p \times 2p}$. Finally, we show that, under our assumption, the dimension of related varieties living on the related ambient space is zero-dimensional.

#### C.2.1    DECOMPOSITION OF ALGEBRAIC QUANTITIES

To potentially identify any latent components of dynamics, we must introduce tensor algebra beyond our current setting, as presented in the following important results.

**Corollary C.1** (CP decomposition). *Let $\mathcal{T} \in \mathbb{R}^{I_1 \times I_2 \times \cdots \times I_N}$ be an order-$N$ tensor. We say that $\mathcal{T}$ admits an exact rank-$R$ Canonical Polyadic (CP) decomposition if there exist component vectors $a_r^{(n)} \in \mathbb{R}^{I_n}$ for each $r = 1, \ldots, R$, $n = 1, \ldots, N$, such that:*

$$\mathcal{T} = \sum_{r=1}^{R} a_r^{(1)} \otimes a_r^{(2)} \otimes \cdots \otimes a_r^{(N)} = [\![A^{(1)}, A^{(2)}, \ldots, A^{(N)}]\!],$$

*where $A^{(n)} = [a_1^{(n)} \; a_2^{(n)} \; \cdots \; a_R^{(n)}] \in \mathbb{R}^{I_n \times R}$ are the factor matrices.*

**Corollary C.2.** *Let $X^{(1)}, X^{(2)}, \ldots, X^{(n)} \in \mathbb{R}^p$ be independent random vectors with nonzero $d$-th order cumulants, such that each admits the form*

$$\kappa_d(X^{(i)}) = \lambda_i \cdot v_i^{\otimes d}, \quad for \; i = 1, \ldots, n,$$

*with $v_i \in \mathbb{R}^p$ and $\lambda_i \in \mathbb{R} \setminus \{0\}$. Let $\mathcal{T} := \kappa_d(X^{(1)} + \cdots + X^{(n)}) \in \mathbb{R}^{p \times \cdots \times p}$ be the $d$-th order cumulant tensor of their sum.*

*Assume that the matrix $V = [v_1 \; v_2 \; \cdots \; v_n] \in \mathbb{R}^{p \times n}$ satisfies*

$$\mathrm{krank}(V) \geq \left\lceil \frac{2n + (d-1)}{d} \right\rceil.$$

*Then the CP decomposition*

$$\mathcal{T} = \sum_{i=1}^{n} \lambda_i \cdot v_i^{\otimes d}$$

*is unique up to scaling and permutation.*

### C.2.2 USEFUL LEMMAS

Recall $F$ is sufficiently generic and $\Phi$ is a kernel matrix with the spectral radius $\rho < 1$. Let $K = F(\mathbb{I}_p - \Phi)^{-1}$ and denote by $\mathcal{F}[\cdot]$ the regular Fourier transformation. We use $[\![K_1, K_2, \ldots, K_p]\!]_{\mathcal{F}}$ to represent the columns of the Fourier-transformed $K$. For each $(K_j)_{\mathcal{F}}$, it has the coordinates $[k_1 : k_2 : \cdots : k_n]_{\mathcal{F}}$ denoting free indeterminates in the vector.

In the sequel, we extend Proposition 4.6 in (Wang & Seigal, 2024) to prove an important pre-identifiability result, the mixed parameter space $F(\mathbb{I}_p - \Phi)^{-1}$.

**Lemma C.1** (Finite intersection with generic linear subspace). *Define a rational map $\psi : \mathbb{P}_{\mathcal{F}}^{n-1} \mapsto \nu_2(\mathbb{P}_{\mathcal{F}}^{n-1}) \subset \mathbb{P}_{\mathcal{F}}^{m-1}$ as:*

$$\psi : [k_1 : k_2 : \cdots : k_n]_{\mathcal{F}} \mapsto [k_1^d : k_1^{d-1} k_2 : \cdots : k_n^d]_{\mathcal{F}} \tag{C.1}$$

*where $m = \binom{n+d-1}{d}$ denotes the ordinates in the projective space of dimension $m-1$. $\nu_d$ represents the $d$-th Veronese embeddings of all order-$d$ tensors. Let $\mathcal{V}$ be the projected space consisting of all rank-1 matrices. Consider a sufficiently generic linear subspace spanned by $\{K_1^{\otimes d}, K_2^{\otimes d}, \cdots, K_p^{\otimes d}\}$, denoted by $\mathcal{W}$. It follows that the variety $\mathcal{W} \bigcap \mathcal{V}$ has dimension zero and $\mathcal{W}$ intersects $\mathcal{V}$ in $d^{n-1}$ distinct points. Therefore, $\{K_1^{\otimes d}, K_2^{\otimes d}, \cdots, K_p^{\otimes d}\}$ is identifiable if and only if $\mathcal{V}(f) = \{K_1^{\otimes d}, K_2^{\otimes d}, \cdots, K_p^{\otimes d}\}$.*

*Proof.* In the classical linear source decomposition (LSD) setting, the $d$-th order cumulant of $X$ admits the following tensor decomposition: $\kappa_d(X) = \sum_{i=1}^{p} \kappa_d(s_i) \cdot (A_i)^{\otimes d}$ under the assumption that the components

of $\epsilon$ are non-Gaussian with non-vanishing $d$-order cumulants, and that multiple interventions are available. The sufficient $d$-order cumulant of each $Z_t$ for a fixed $t = t_i$ is

$$\kappa_d(Z_t) = \kappa_d[(I - \Phi)^{-1} \star \epsilon_t] = \kappa_d[\sum_{k=1}^{t} H_{t-s}\epsilon_s]$$

For each $s$, the linear transformation $H_{t-s}$ results in a multi-linear transformation of their cumulants

$$\kappa_d(H_{t-s}\epsilon_k) = (H_{t-s})^{\otimes d}\mathcal{C}_{\epsilon_s}^d = \sum_{i=1}^{p} \kappa_d(\epsilon_s^i)(H_{t-s})_j^{\otimes d}$$

The full $d$-order cumulant is

$$\kappa_d(O_t) = \kappa_d(\underbrace{FZ_t, FZ_t, \ldots, FZ_t}_{d \text{ times}})$$

$$= F^{\otimes d} \cdot \kappa_d(\underbrace{Z_t, Z_t, \ldots, Z_t}_{d \text{ times}}) \tag{C.2}$$

$$= \underbrace{F \otimes F \otimes \cdots \otimes F}_{d \text{ times}} \cdot \kappa_d \left( \underbrace{\sum_{s_1=1}^{t} H_{t-s_1}\epsilon_{s_1}, \sum_{s_2=1}^{t} H_{t-s_2}\epsilon_{s_2}, \ldots, \sum_{s_d=1}^{t} H_{t-s_d}\epsilon_{s_d}}_{d \text{ times}} \right)$$

$$= \underbrace{F \otimes F \otimes \cdots \otimes F}_{d \text{ times}} \cdot \sum_{s_1=1}^{t} \cdots \sum_{s_d=1}^{t} \kappa_d \left( H_{t-s_1}\epsilon_{s_1}, H_{t-s_2}\epsilon_{s_2}, \ldots, H_{t-s_d}\epsilon_{s_d} \right)$$

$$= \underbrace{F \otimes F \otimes \cdots \otimes F}_{d \text{ times}} \cdot \sum_{s_1=1}^{t} \cdots \sum_{s=1}^{t} \sum_{j=1}^{p} \kappa_d \left( H_{t-s}^{(:,j)}\epsilon_s^{(j)}, H_{t-s}^{(:,j)}\epsilon_s^{(j)}, \ldots, H_{t-s}^{(:,j)}\epsilon_s^{(j)} \right)$$

$$= \underbrace{F \otimes F \otimes \cdots \otimes F}_{d \text{ times}} \cdot \left( \sum_{s=1}^{t} \sum_{j=1}^{p} \kappa_d^{(j)}(\epsilon) \cdot \underbrace{H_{t-s}^{(:,j)} \otimes H_{t-s}^{(:,j)} \otimes \cdots \otimes H_{t-s}^{(:,j)}}_{d \text{ times}} \right)$$

$$= \left( \sum_{s=1}^{t} \sum_{j=1}^{p} \kappa_d^{(j)}(\epsilon) \cdot \underbrace{F \otimes F \otimes \cdots \otimes F}_{d \text{ times}} \cdot \left( H_{t-s}^{(:,j)} \right)^{\otimes d} \right)$$

$$= \sum_{s=1}^{t} \sum_{j=1}^{p} \kappa_d^{(j)}(\epsilon) \cdot \left( FH_{t-s}^{(:,j)} \right)^{\otimes d} \tag{C.3}$$

We drop the time index in $\kappa(\epsilon)$ since the noise is temporally uncorrelated (white noise), i.e.,

$$\mathrm{Cov}(\epsilon_s, \epsilon_t) = \sigma^2 \delta_{s,t},$$

Unlike in a time-free process, the joint cumulant of a time process is of order $d$ that is coupled with the number of time lags:

$$\kappa_d(O_{t_1}, \ldots, O_{t_d}) = \sum_{s=1}^{\min(t_1, \ldots, t_d)-1} \sum_{j=1}^{p} \kappa_d^{(j)}(\epsilon) \cdot \bigotimes_{\ell=1}^{d} \left( FH_{t_\ell - s}^{(:,j)} \right)$$

$$= \sum_{j=1}^{p} \sum_{s=1}^{\min(t_1,\ldots,t_d)-1} \kappa_d^{(j)}(\epsilon) \cdot \bigotimes_{\ell=1}^{d} \left( FH_{t_\ell-s}^{(:,j)} \right) \tag{C.4}$$

We denote the Fourier transform of $x(t)$ with respect to time $t$ as $\mathcal{F}[x](\omega)$. Using the convolution theorem and linearity of the Fourier transform, we have:

$$\mathcal{F}[\kappa_d(O_t)](\omega) = \mathcal{F}\left[ \sum_{s\geq 0}^{t} \sum_{j=1}^{p} \kappa_d^{(j)}(\epsilon) \cdot \left( FH_{t-s}^{(:,j)} \right)^{\otimes d} \right]$$

$$= \mathcal{F}\left[ \int_0^t \sum_{j=1}^{p} \kappa_d^{(j)}(\epsilon) \cdot \left( FH(t-s)^{(:,j)} \right)^{\otimes d} ds \right]$$

$$= \left( \sum_{j=1}^{p} \kappa_d^{(j)}(\epsilon) \cdot \mathcal{F}\left[ \left( FH^{(:,j)} \right)^{\otimes d} \star \mathcal{U}(t-s) \right] \right)$$

$$= \left( \sum_{j=1}^{p} \kappa_d^{(j)}(\epsilon) \cdot \mathcal{F}\left[ \left( FH^{(:,j)} \right)^{\otimes d} \right](\omega) \cdot \left( \pi\delta(\omega) + \frac{1}{i\omega} \right) \right) \tag{C.5}$$

Hereafter, we applied $\mathcal{U}$ to induce a generalized convolution integral. Since $\delta(\omega)$ vanishes everywhere except at $\omega = 0$, multiplying it by $\omega$ gives zero, Eq. (C.5) yields

$$i\omega \mathcal{F}[\kappa_d(O_t)](\omega) = \left( i\omega \sum_{j=1}^{p} \kappa_d^{(j)}(\epsilon) \cdot \mathcal{F}\left[ \left( FH^{(:,j)} \right)^{\otimes d} \right](\omega) \cdot \left( \pi\delta(\omega) + \frac{1}{i\omega} \right) \right)$$

$$= \left( \sum_{j=1}^{p} \kappa_d^{(j)}(\epsilon) \cdot \mathcal{F}\left[ \left( FH^{(:,j)} \right)^{\otimes d} \right](\omega) \cdot i\omega \cdot \left( \pi\delta(\omega) + \frac{1}{i\omega} \right) \right).$$

which reads

$$\left( \sum_{j=1}^{p} \kappa_d^{(j)}(\epsilon) \cdot \mathcal{F}\left[ \left( FH^{(:,j)} \right)^{\otimes d} \right](\omega) \cdot i\omega \cdot \left( \pi\delta(\omega) + \frac{1}{i\omega} \right) \right) = \left( \sum_{j=1}^{p} \kappa_d^{(j)}(\epsilon) \cdot \mathcal{F}\left[ \left( FH^{(:,j)} \right)^{\otimes d} \right](\omega) \cdot 1 \right) \tag{C.6}$$

Writing $FH$ in terms of $K$ as a convention obtains:

$$\kappa_{O_t,\mathcal{F}} = \left( \sum_{j=1}^{p} \kappa_d^{(j)}(\epsilon) \cdot (k_j)_{\mathcal{F}}^{\otimes d} \right)$$

$(k_j)_{\mathcal{F}}^{\otimes d}$ is an order $d$, rank 1 tensor and thus can be represented as the space of $V \otimes \cdots \otimes V$. In projective space $\mathbb{P}^n$, each point represents a line going through the origin, and all points lose one dimension up to multiples. Therefore, the variety $\mathcal{V}$ has projective dimension $n-1$, embedded in an ambient space of projective dimension $m-1$. The linear subspace $\mathcal{W}$ is an element of the Grassmannian $\mathrm{Gr}(p-1,n-1)$ and is sufficiently generic. By Proposition 2.6 of (Chiantini & Ciliberto, 2002), for a sufficiently generic, reduced, irreducible variety $\mathcal{V}$ in $\mathbb{P}^{m-1}$, if there exists an integer such that $k + (n-1) < m-1$, then the intersections of $\mathcal{V}$ with the generic linear subspace $\mathcal{W}$ are the union of the generic points $\langle P_0, P_1, \cdots, P_k \rangle$.

We can prove this by choosing $k = p$. This condition can be much more easily satisfied if we have a higher order $d \geq 2$ due to the combinatorial nature of $m = \binom{n+d-1}{d}$.

Consequently, by assuming non-Gaussianity in $\epsilon_t$, for each $j$, Eq. (C.6), hence $i\omega\mathcal{F}[\kappa_d(O_t)](\omega)$ has a unique decomposition of the summation of a rank-1 tensor(matrix if $d = 2$). Therefore, each column of the sub-linear mixing transferring matrix $\mathcal{F}\left[\left(FH^{(:,j)}\right)\right]$ is theoretically recovered up to a scaling and permutation $\pi$ if all assumptions made are satisfied for $\Phi$. This indicates that, regardless of whether the decomposition must be explicitly calculated, such uniqueness provides a foundation for further disentanglement. $\qquad\square$

Hereafter, $\mathcal{F}\left[\left(FH^{(:,j)}\right)\right] DP$ is available; we thus obtain the unique indeterminacy as an immediate result of Lemma. C.2.

**Lemma C.2.** *Consider $\mathbb{F}$ is an algebraically closed field, the unknown indeterminacy $DP$ is preserved in $\mathbb{F}$, that is, the following relation*

$$FH_\tau^{(:,j)} = \hat{F}\hat{H}_\tau^{(:,j)} D_{j,\tau} P_{j,\tau},$$

$$with \ (P_{j,\tau}, D_{j,\tau}) \in \{(P_j, D_j) \mid \mathcal{F}[FH^{(:,j)}] D_j P_j = \mathcal{F}[\hat{F}\hat{H}^{(:,j)}]\}.$$

*Proof.* Let $\mathbb{F}$ be a field and $n \geq 1$ an integer. The *general linear group* of degree $n$ over $\mathbb{F}$ is

$$\mathrm{GL}_n(\mathbb{F}) := \{ A \in M_n(\mathbb{F}) \mid \det(A) \neq 0 \}.$$

Equivalently, if $V$ is a $n$-dimensional vector space over $\mathbb{F}$,

$$\mathrm{GL}(V) := \mathrm{Aut}_{\mathbb{F}}(V) = \{ T : V \to V \text{ linear isomorphisms} \},$$

and any choice of basis identifies $\mathrm{GL}(V)$ with $\mathrm{GL}_n(\mathbb{F})$. It is obvious that $\mathcal{F}[\mathbb{F}]$ is exactly a subgroup of $\mathrm{GL}(\mathbb{F})$ as the group $\mathrm{GL}_n(\mathbb{F})$ satisfies: i) It is precisely the set of all invertible linear transformations (invertible matrices). ii) If $\mathbb{F} = \mathbb{R}$ or $\mathbb{C}$, then $\mathrm{GL}_n(\mathbb{F})$ is an open subset of $M_n(\mathbb{F})$ since $\mathrm{GL}_n(\mathbb{F}) = \det^{-1}(\mathbb{F} \setminus \{0\})$, and it is a Lie group. By the definition of kernel matrix, one notes that i) trivially holds due to the maximal spectrum being less than 1. For ii), $\det^{-1}$ denotes the preimage of the open set $\mathbb{F} \setminus \{0\}$ under $\det$. Cutting the one-dimensional line at 0 produces two open intervals (for $\mathbb{F} = \mathbb{R}$) or a punctured plane (for $\mathbb{F} = \mathbb{C}$), hence the preimage is open in $M_n(\mathbb{F})$. Therefore, the permutation and scaling must be preserved in $M \in \mathbb{R}^{p \times p}$. $\quad\square$

Using Lemma C.2, the generic points $(k_j)_{\mathcal{F}}^{\otimes d}(w)$ with a generic projective linear span indicate generic points $(k_j)^{\otimes d}(\tau)$. As a result, the original kernel mixing matrix $FH_\tau$ is recovered up to the same permutation and scaling for any $\tau$. In what follows, the claim to be established is the recovery of the causal structure as well as its full parameter space. Our proof focuses on the polynomial system and its associated ideal $\mathcal{I}$ generated by the multi-linear constrained polynomial system.

### C.2.3 PROOF OF THEOREM 1

What remains to be proved is to show that, in Theorem 1 , condition (2) ensures condition (3), which guarantees the full identifiability of the model $\Theta := (F, \Phi, U)$. To study the geometry of the parameter space of the latent model, we endow a topological ordering to $\Phi$.

**Lemma C.3.** *Given a bipartite graph of the proposed INAR($\infty$) structure, it admits a* kernel DAG, *denoted by $\mathscr{G}$, corresponding to a matrix $\mathcal{M}_{\mathscr{G}} \in \mathbb{R}^{2p \times 2p}$ such that $\mathbb{I}_{2p} - \mathcal{M}_{\mathscr{G}}$ is invertible. Consequently, its inverse can be expressed as a finite order $k$ expansion of $\mathcal{M}_{\mathscr{G}}$,*

$$(\mathbb{I}_{2p} - \mathcal{M}_{\mathscr{G}})^{-1} = \sum_{i=0}^{k} \mathcal{M}_{\mathscr{G}}^i, \quad k \leq p, \quad \mathcal{M}_{\mathscr{G}} = \begin{bmatrix} \mathcal{M}_{\mathscr{G}}[U] & \Phi \\ 0 & \mathcal{M}_{\mathscr{G}}[V] \end{bmatrix}$$

*where $k$ corresponds to the length of the longest path in the DAG and $\mathcal{M}_{\mathscr{G}}[U] = \mathcal{M}_{\mathscr{G}}[V] = \mathbf{0}_{p \times p}$.*

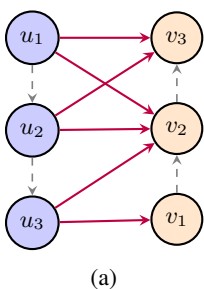

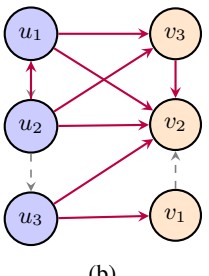

(a)                                                                 (b)

Figure A3: Figure(a) is a kernel-delayed DAG $\mathscr{G}$ compatible with $\mathcal{M}_{\mathscr{G}}$; the system $\mathcal{M}_{\mathscr{G}}$ is a strict upper-triangular matrix. Figure (b) violates the topological order with additional edges $\{u_2 \to u_1, v_3 \to v_2\}$.

*Proof.* Let $\mathcal{M}_{\mathcal{G}} \in \mathbb{R}^{2p \times 2p}$ be the kernel matrix associated with the bipartite graph, where the variables are partitioned into two subsets $U$ and $V$. Consider a topological ordering where all nodes in $U$ precede those in $V$. Since edges from $V$ to $U$ are forbidden by the bipartite structure, no element in the lower-left block of $\mathcal{M}$ can be nonzero. Moreover, edges within $U$ are only topologically ordered, so the off-diagonal and upper-right block corresponding to $U$ have zeros on the diagonal. The edges within $V$ form a DAG, and under a topological ordering of $V$, the corresponding block in $\mathcal{M}$ is strictly upper-triangular. Therefore, $\mathcal{M}$ as a whole is strictly upper-triangular, which implies that it represents a DAG.

Because $\mathcal{M}_{\mathcal{G}}$ is strictly upper-triangular, it is nilpotent. Let $k$ denote the length of the longest directed path in the DAG. Then $\mathcal{M}_{\mathcal{G}}^{k+1} = 0$, and the inverse of $\mathbb{I} - \mathcal{M}_{\mathcal{G}_K}$ can be expressed as a finite sum of powers of $\mathcal{M}$:

$$(\mathbb{I} - \mathcal{M}_{\mathcal{G}})^{-1} = \sum_{i=0}^{k} \mathcal{M}_{\mathcal{G}}^{i}.$$

Each term $\mathcal{M}_{\mathcal{G}}^{i}$ corresponds to contributions from paths of length $i$ in the DAG. This shows that the inverse is fully determined by path products up to the longest path length $k$, completing the proof. $\square$

*Example* 2. Consider a kernel matrix $\Phi \in \mathbb{R}^{2 \times 2}$ with internal arrows allowed:

$$\Psi_U = \begin{bmatrix} 0 & \psi_{12} \\ 0 & 0 \end{bmatrix}, \quad \Phi = \begin{bmatrix} \phi_{11} & \phi_{12} \\ \phi_{21} & \phi_{22} \end{bmatrix}, \Psi_V = \begin{bmatrix} 0 & \psi'_{34} \\ 0 & 0 \end{bmatrix}$$

where $\Psi_U, \Psi_V$ encodes the internal arrows of the sub-graph $\mathscr{G}_U$ and $\mathscr{G}_V$; $\Phi$ encodes time-delayed kernel effects from $s$ to $t$.

The corresponding expanded kernel matrix $\mathcal{M} \in \mathbb{R}^{4 \times 4}$ is

$$\mathcal{M} = \begin{bmatrix} 0 & \psi_{12} & \phi_{11} & \phi_{12} \\ 0 & 0 & \phi_{21} & \phi_{22} \\ 0 & 0 & 0 & \psi'_{34} \\ 0 & 0 & 0 & 0 \end{bmatrix}.$$

The polynomial system in condition (3) has a geometric equivalence representation. That is, condition (3) immediately indicates Corollary C.3.

**Corollary C.3.** *The ideal* $I : \langle K_{\mathscr{G}}(\mathbb{I}_{2p} - M_{\mathscr{G}}) - F_{\mathscr{G}} \rangle$ *is such that the space* $\Theta_{\mathscr{G}} := (F_{\mathscr{G}}, M_{\mathscr{G}})$ *is zero-dimensional.*

By Corollary C.3, identifying the space of all parameters in $F$ and $\Phi$ is equivalent to solving the following ideal $\mathcal{I} : \langle F_{\mathscr{G}} - F_{\mathscr{G}}(\mathbb{I}_{2p} - \mathcal{M}_{\mathscr{G}})^{-1}(\mathbb{I}_{2p} - \mathcal{M}_{\mathscr{G}}) \rangle$. In latent causal models, $F_{\mathscr{G}}, \mathcal{M}_{\mathscr{G}}$ are filled as indeterminates

that need to be recovered, where $F_{\mathscr{G}}$ is the expanded linear mixing obtained by filling $F \in \mathbb{R}^{n \times p}$ in a larger block-diagonal matrix of size $2n \times 2p$, denoted by $F_{\mathscr{G}}$.

Remember, we have $F(\mathbb{I} - \Phi(\tau))^{-1}$ to be unique due to decomposition up to a scaling and permutation. We write $H_{\mathscr{G}} = (\mathbb{I}_{2p} - \mathcal{M}_{\mathscr{G}})^{-1}$, then

$$F_{\mathscr{G}}(\mathbb{I}_{2p} - \mathcal{M}_{\mathscr{G}_K})^{-1} = \begin{bmatrix} F & \mathbf{0} \\ \mathbf{0} & F \end{bmatrix} \begin{bmatrix} H_{\mathscr{G}}(U) & H_{\mathscr{G}}(\Phi) \\ \mathbf{0} & H_{\mathscr{G}}(V) \end{bmatrix} = \begin{bmatrix} FH_{\mathscr{G}}(U) & FH(\Phi) \\ \mathbf{0} & FH_{\mathscr{G}}(V) \end{bmatrix} = K_{\mathscr{G}}$$

Under INAR, the diagonal blocks of $K_{\mathscr{G}}$ are $\mathbf{0}$ matrix. Therefore, we have the ideal $\mathcal{I} : \langle F_{\mathscr{G}} - K_{\mathscr{G}}(\mathbb{I}_{2p} - \mathcal{M}_{\mathscr{G}}) \rangle$ where $K_{\mathscr{G}}$ is known because $FH$ is known, by Lemma C.1, so they are no longer considered as indeterminate and thus do not contribute any degrees in the dimension of $\mathcal{I}$. Clearly, under passively observational settings, recovery of full models is never possible as the current $\mathcal{I}$ must be positive dimensional, leading to no fixed points defined in the associated variety $\mathcal{V}$. This leads to a central goal of identifying the number of contexts that indicate sufficient variability or interventional settings, thereby recovering the parameter space. To this end, we need to first discuss important properties of $F_{\mathscr{G}}$.

**Lemma C.4.** *$F \in \mathbb{R}^{n \times p}$ is a generic full-rank matrix. Then $F_{\mathscr{G}}$ is of full rank with $\text{rank}(F_{\mathscr{G}}) = 2 \cdot \text{rank}(F) = 2\min(n, p)$, and it is not generic in an open dense subset of $\mathbb{R}^{2n \times 2p}$ due to the additional linear constraints imposed by the block-diagonal structure. Consequently, $F_{\mathscr{G}}$ belongs to a proper linear subvariety of $\mathbb{R}^{2n \times 2p}$ defined by*

$$\mathcal{V}_{I_\star} := \left\{ B \in \mathbb{R}^{2n \times 2p} : B = \begin{pmatrix} F' & 0 \\ 0 & F' \end{pmatrix}, \ F' \in \mathbb{R}^{n \times p} \right\}.$$

*Therefore, for any square matrix $A^{2p}$ with $\text{rank}(A) \leq r$, $\text{rank}(F_{\mathscr{G}} A) \leq r$*

*Proof.* This proof is trivial by linear algebra. $\square$

We first show that identifying $\mathcal{M}$ is equivalent to making a variety, which falls onto a projective space $\mathbb{P}^d$ that is zero-dimensional. However, as for the current ideal, the dimension can be rather larger, as shown in the following lemma.

**Lemma C.5.** *The subvariety associated with the ideal $\mathcal{I}^\star$ for $\mathcal{M}_{\mathscr{G}}$ has dimension at least $3p^2 - 2p$.*

*Proof.* Consider the matrix $\mathcal{M}_{\mathscr{G}}$ in the ambient projective space. Since the bottom-left block $\mathcal{M}_{\mathscr{G}}$ is forced to be zero, we have $\mathcal{M}_{\mathscr{G}} \in \mathbb{P}^{4p^2 - 1 - p^2}$. $\mathcal{M}_{\mathscr{G}}$ is nilpotent, i.e., $\mathcal{M}_{\mathscr{G}}^k = 0$. This nilpotency imposes additional algebraic constraints, restricting $\mathcal{M}_{\mathscr{G}}$ to a subvariety $V^\star = \langle \text{Tr}(\mathcal{M}_{\mathscr{G}}), \det(\mathcal{M}_{\mathscr{G}}) \rangle$. The dimension of this variety is

$$\dim V^\star = (2p)^2 - 1 - (2p) = 4p^2 - 2p - 1.$$

Now, the intersection of two varieties of dimensions $d_1$ and $d_2$ in an ambient space of dimension $n$ satisfies

$$\dim(V_1 \cap V_2) \geq d_1 + d_2 - n.$$

Applying this to our case, we have

$$\dim(\text{variety satisfying all constraints}) \geq (4p^2 - 1 - p^2) + (4p^2 - 2p - 1) - 4p^2 = 3p^2 - 2p.$$

Hence, the subvariety associated with $\mathcal{I}^\star$ has dimension at least $3p^2 - 2p$. $\square$

As shown in Lemma C.5, we need at least extra $3p^2 - 2p - 1$ independent polynomials to cut off $\mathcal{V}$ that identify all $2p \times 2p$ augmented kernel matrices.

Note we have $p$ processes; we generally assume we obtain different contextual information from at least $K$ environments (i.e., $K = p$), which is a mild condition in causal representation learning. Each sub-ideal $\mathcal{I}_k$ : $\langle F_{\mathcal{G}} - K_{\mathcal{G}}(\mathbb{I}_{2p} - \mathcal{M}_{\mathcal{G}}) \rangle$ constitutes a polynomial system, denoted by $\mathcal{S}_{\mathbb{P}}^k$ such that:

$$F_{\mathcal{G}} + K_{\mathcal{G}}^{(k)} \mathcal{M}_{\mathcal{G}}^{(k)} = K_{\mathcal{G}}^{(k)}. \tag{C.7}$$

There are $2n \times 2p$ indeterminates for $F_{\mathcal{G}}$ and $2|e(\mathcal{G})|$ for $\mathcal{M}$ since each environment $k$ introduces a new $\mathcal{M}_{k,j}$. Considering all $K$ ideals $(\mathcal{I}_0, \mathcal{I}_2, \cdots, \mathcal{I}_K)$, we obtain the union of all $K$ varieties:

$$V(\mathcal{I}) = \left\{ (F_{\mathcal{G}}, \mathcal{M}_{\mathcal{G}}^{(0)}, \ldots, \mathcal{M}_{\mathcal{G}}^{(K)}) \,\Big|\, F_{\mathcal{G}} - K_{\mathcal{G}}^{(k)} (\mathbb{I}_{2p} - \mathcal{M}_{\mathcal{G}}^{(k)}) = 0, \ \forall k \in K \right\}. \tag{C.8}$$

Adding polynomial constraints by subtracting Eq.(C.7) for 0 from that for $k$ obtains:

$$V(\mathcal{I}^\star) = \left\{ V(\mathcal{I}) \,\Big|\, K_{\mathcal{G}}^{(k)} \mathcal{M}_{\mathcal{G}}^{(k)} - K_{\mathcal{G}}^{(0)} \mathcal{M}_{\mathcal{G}}^{(0)} - (K_{\mathcal{G}}^{(k)} - K_{\mathcal{G}}^{(0)}) = 0, \ \forall k \in K \right\}. \tag{C.9}$$

that induces a coordinate ring $R/\mathcal{I}$ in a polynomial ring $R = k(F_{\mathcal{G}_{i,j}}, \mathcal{M}_{i,j}^k)$. The order of the coordinate ring is the dimension of the original variety. We use $\left( -\!\!\mid\!\!- \right)$ to represent the blocked system so as not to let readers confuse it with the matrix bracket. For each $m \in [n], j \in [p]$, and $i \in \mathrm{ch}_{\mathcal{G}}(j)$, we have $|\mathrm{ch}_{\mathcal{G}}(j)|$ columns for $\mathcal{M}_{\mathcal{G}}$. $F_v, \mathcal{M}_v^{(k)}$ write entries $f_{i,j}, \mathcal{M}_{m,i}^k$ as vectors, which describe the polynomial constraints as a linear system:

$$\left( \begin{array}{c|c} \mathbb{I}_{4np} & \star \\ \hline \mathbf{0} & \star \end{array} \right) \begin{pmatrix} F_v \\ \mathcal{M}_v^{(0)} \\ \mathcal{M}_v^{(k)} \end{pmatrix} = \left( \frac{K_{\mathcal{G}}^{(k)}}{K_{\mathcal{G}}^{(k)} - K_{\mathcal{G}}^{(0)}} \right). \tag{C.10}$$

In INAR $(\infty)$, each $\mathcal{M} := \mathcal{M}_{t-s}$ preserves all paths $\{(\varphi(j) \to \varphi(i)|Z_{\varphi(j),t-s}^\Delta \to Z_{\varphi(i),t}^\Delta, \mathcal{M}_{i,j} \neq 0\}$ from $\Phi_{t-s}$ through an isomorphism $\varphi$. Therefore, entry $(\mathbb{I}_{2p} - \mathcal{M})_{i,j}^{-1}$ is the product of $\mathcal{M}_{n,m}$ for path $j \to m \to n \to i$. We drop the time index and graph label whenever the context is clear. Such $(\mathbb{I}_{2p} - \mathcal{M})_{i,j}^{-1}$ admits the representation

$$(\mathbb{I}_{2p} - \mathcal{M})^{-1} = \mathbb{I} + \mathcal{M}.$$

Results from (Carreno et al., 2024) are applied to get $\mathrm{rank}(\mathbb{I}_{2p} - \mathcal{M}^{(k)})^{-1} - (\mathbb{I}_{2p} - \mathcal{M}^0)^{-1} \leq 1$. Using Lemma C.4, we obtain $\mathrm{rank}(F_{\mathcal{G}}(\mathbb{I}_{2p} - \mathcal{M}^{(k)})^{-1} - F_{\mathcal{G}}(\mathbb{I}_{2p} - \mathcal{M}^0)^{-1}) \leq 1$. Therefore, the left part of Eq.(C.10) has columns that are multiples of each other:

$$(K_{\mathcal{G}}^{(k)} - K_{\mathcal{G}}^{(0)})_{l,j} = (K_{\mathcal{G}}^{(0)})_{l,k} \Delta_{k,i}, \quad \Delta := (I - \mathcal{M}^{(k)})^{-1} - (I - \mathcal{M}^{(0)})^{-1} \tag{C.11}$$

We examine the non-zero sub-blocks of the lower-right block $\star$ in Eq.(C.10), which has size $|\mathrm{de}(j) \setminus \mathrm{ch}(j)| \times |\mathrm{ch}(j)|$. Following the convention, we represent the sunblocks as $M[j]$ and choose the smaller blocks $[K_{\mathcal{G}}^{(k)} - K_{\mathcal{G}}^{(0)}]$ corresponding to the size of $M[j]$ and write it as $b[j]$. The dimension of variety $V(\mathcal{I}^\star)$ is the dimension of the points $(\mathcal{M}_{i,j}^0)$, $i \in \mathrm{ch}(j)$ that satisfy:

$$M[j](\mathcal{M}_{i,j}^0) = b[j] \tag{C.12}$$

The variety is a null set when the above constraints lead to no solutions. Therefore, we require $\mathrm{rank}(M[j]) = \mathrm{rank}(M[j]|b[j])$. $[M[j] \mid b[j]]$ is the common augmented matrix to check the stability of a polynomial equation system. We conclude our proof by making a formal statement about the dimension of $\mathcal{V}(\mathcal{I}^\star)$ in the next lemma.

**Lemma C.6.** *For generic $F$ and $FH$ arising from the cumulant decomposition, the full generating model is identifiable if and only if the variety $\mathcal{V}(\mathcal{I}^\star)$ has dimension zero, that is,*

$$\dim(\mathcal{V}(\mathcal{I}^\star)) = \sum_{j=1}^q \mathrm{ch}(j) - \mathrm{rank}(M[j]) = 0.$$

*Proof.* For the left subset $U_s$, each node $j$ has outgoing edges only to nodes $i$ in the right subset $V_t$, all of which are direct children of $j$. By construction, no edges exist within $U_s$ or within $V_t$. Consequently, for each $j$, we have $M[j] = \emptyset$, since $\mathrm{de}(j) = \mathrm{ch}(j)$. $\qquad\square$

When $\mathcal{M}_{t-s}$ is fully recovered, the full matrix $\mathcal{F}_\mathscr{G}$ can be obtained as

$$\mathcal{F}_\mathscr{G} = K_\mathscr{G}(\mathbb{I}_{2p} - \mathcal{M}_\mathscr{G}).$$

If $F_v$ is injective, identification of $F_v$ (and hence $\mathcal{F}_\mathscr{G}$ and $F$) is equivalent to identifying each $\mathcal{M}$ individually, due to the direct multiplication of $F^{-1}$ (or the pseudo-inverse $F^\dagger$) with $K$. However, identification of the full generating model is hindered by the genericity of $F$. Even if $K_\mathscr{G}$ is unique up to the usual indeterminacies, recovering other kernel matrices $\mathcal{M}_{t-s'}$ requires analogous identifiability conditions for the individual kernel matrix. It follows that the identifiability of the model is governed by the geometric complexity of its embedding. In particular, at least $p$ generic points in the ambient projective space $\mathbb{P}^n$ are required. If such generic points are obtained solely by varying the parameter $\phi$, then the resulting points lie in the same projective orbit

$$\mathcal{O}_\phi \;=\; \{[\Phi(\phi + \delta)] \;\mid\; \delta \in \mathbb{R}^k\}$$

under parameter shifts. Consequently, identifiability holds only up to this equivalence, that is,

$$\phi_1 \sim \phi_2 \quad \Longleftrightarrow \quad \Theta(\Phi_{i:}(\phi_1)) = \Theta(\Phi_{i:}(\phi_2)),$$

so that distinct parametrizations of the same process become indistinguishable whenever they correspond to pairs of identical parameter rows $\Theta(\Phi_{i:})$. Therefore, in continuous time, the process has a natural shift as long as $\Theta(\Phi_{i:})$ is changed. Hence, under $k \in 1, 2, ..., K$ distinct contexts—each introducing sufficient variability in the distribution, or ensuring that each lag $k$ receives at least one intervention that shifts the downstream mechanism—the full latent structure is identifiable up to the same indeterminacy.

**Recovery of baseline $U$.** Once the full causal structure is recovered up to a scaling and permutation matrix, lower level moments of $\mathbb{E}[F(\mathbb{I}_p - \Phi)^{-1} \star (U + \epsilon_t)]$ are can be directly computed to find $U$ up to the same indeterminacy.

**Remark for conditions.** White noise ensures all components of the process $\epsilon_t$ are mutually independent, causing the integral in line with $\epsilon(\Delta)$ the Fourier transformation to be absolutely zero except for $\Delta = 0$, and leaves the algebraic cumulant $\kappa(O_t)$ with a reduced, parsable form that admits a unique decomposition. We highlight a case when the proposed condition is seriously violated if a random Wiener process $dW_t$ is chosen in place of a white perturbation. Therefore, identifying a completely random stochastic differential process remains challenging since the increment-independent perturbation still forms a time-dependent noise after the Ito integral is applied. For completeness, in C.5.1 , we show how dependent noise can be reduced to independent-increment noise while keeping the identifiability.

### C.2.4 IDENTIFIABILITY UNDER GAUSSIAN NOISE

Now we focus on the case where non-Gaussianity does not hold for the entire time process. When the noise is Gaussian, $\kappa_d(O_t) = 0$ for all $d \geq 3$, which leads to a $\mathbf{0} \in \mathbb{R}^{p \times d}$ (the $d$-th order zero tensor). The solution of decomposition is infinite, thus $FH$ cannot be recovered up to a column scaling and permutation, nor can the latent transition graph $\mathcal{G}$. We argue that preserving only $d$- order cumulant of order $d \leq 2$ is a minimal building block for identification. $\kappa_2(O_t)$ is an order 2 variance-covariance matrix.

The defining ideal $I(\mathcal{V})$ is generated by all $2 \times 2$ minors. The defining ideal $I(\mathcal{V})$ is generated by all $2 \times 2$ minors. More generally, explicit generators include

$$k_{ij}k_{k\ell} - k_{i\ell}k_{kj} = 0 \quad (1 \leq i, j, k, \ell \leq n).$$

Geometrically, $\mathcal{V}$ satisfies $\dim \mathcal{V} = n - 1$ and it is the quadratic Veronese variety of degree $2^{n-1}$.

Let $M_1, M_2, \ldots, M_T \in \mathbb{R}^{p \times p}$ be a collection of order-2 tensors. We define the order-3 tensor $\mathcal{X} \in \mathbb{R}^{T \times p \times p}$ via concatenation along the first mode (tensor slices):

$$\mathcal{K}_c = \mathrm{con}\,(M_1, M_2, \ldots, M_T), \quad \text{where } \mathcal{X}_{t,:,:} = M_t. \tag{C.13}$$

By standard tensor algebra, an order-3 tensor $\mathcal{X} \in \mathbb{R}^{T \times p \times p}$ can be reshaped or flattened into a higher-order tensor, under a specific indexing scheme. More generally, given a desired tensor order $d$, and assuming $T = p^{d-2}$, we define a transformation:

$$\mathcal{T} : \mathbb{R}^{T \times p \times p} \to \mathbb{R}^{p^d}, \; \kappa_d(\varepsilon_t) \in \mathbb{R}^{\overbrace{p \times \cdots \times p}^{d \text{ times}}}$$

$$\kappa_2(\varepsilon_t) = \begin{bmatrix} \sigma_{11} & \sigma_{12} & \cdots & \sigma_{1p} \\ \sigma_{21} & \sigma_{22} & \cdots & \sigma_{2p} \\ \vdots & \vdots & \ddots & \vdots \\ \sigma_{p1} & \sigma_{p2} & \cdots & \sigma_{pp} \end{bmatrix} \in \mathbb{R}^{p \times p}, \kappa_3(X_t) = \begin{bmatrix} \mathbf{0}_{p \times p} \\ \mathbf{0}_{p \times p} \\ \vdots \\ \mathbf{0}_{p \times p} \end{bmatrix} \in \mathbb{R}^{p \times p \times p}$$

that re-indexes the tensor slices $M_t$ to fill the missing indices of an order $d$ cumulant tensor. The replacing and re-indexing rule is illustrated in an order 3 tensor as a real plane, assuming $d - 1$ is the maximal order such that $\kappa_d = 0$.

Under this transformation, each slice $M_t$ is interpreted as contributing to a specific mode configuration of the higher-order tensor. That is, the tensor $\mathcal{X}$ is "lifted" into a $d$-way tensor by embedding each $p \times p$ matrix slice as filling in the cumulant entries with fixed positions in the first $d - 2$ indices corresponding to $t \in \{1, \ldots, T\}$, and varying the remaining two indices over $p \times p$. This leads to the same form as Eq. (C.4) where all $\kappa_d(\epsilon_t) \neq 0$. Under Corollary C.1 and C.2, the new tensor has a unique decomposition of rank-1 tensor summation. To be specific, we assume $v_i$ has no pair of columns to be collinear. This ensures the identification of $F(I - \Phi)^{-1}$ and restricts $F$ to be injective to only $\mathrm{span}(H_j)$.

The sequential steps are the same for non-Gaussian noise since the construction of the ideal $\mathcal{I}^\star$ associated with its variety is not influenced by $\epsilon$ once $FH$ is fixed up to a permutation and scaling.

**Discussion of noise.** Consequently, when Gaussianity is assumed, the distribution of $O_t$ is fully characterized by the first two cumulants. This property implies that the entire cumulant expansion, and hence any higher-order dependency, collapses at second order. In this sense, the Gaussian distribution is the unique fixed point of the cumulant hierarchy at order two. In temporal parametric transition (Yao et al., 2022b), a widely known condition to ensure the component-wise identifiability of the latent process $Z_t$ is to require that the driving noise $\epsilon$ is not a fully isotropic Gaussian. That is, the Gaussian noise distribution must shift under either intervention (Buchholz et al., 2023) or exhibit heterogeneity in its variance. This is because all cumulants of order $p > 2$, which encode the exact causal dependencies, vanish for Gaussian noise. As a result, sufficient variability can only arise from changes in the second-order cumulant. Our results reflect that non-isotropic Gaussian noise is a *necessary* but not a *sufficient* condition for full identifiability of the time-delayed generative model and parameters.

### C.2.5 Identifying causal structure

Our proof is constructive: Since our results also apply to any soft interventions with the minimal manifold $\kappa_d(O_t)$ such that no fixed value of entry $(i \to j)$ in $\Phi(w)$ induces a dependence removal: Without loss

of generality, we can safely choose the kernel matrix of order $p$ as weak convergence exists. Therefore, we discretize the continuous kernel matrix up to $t - s = \tau$, each $\tau \in (\tau_1, \cdots, \tau_p)$. Applying the Fourier transformation to the above equation for order $p$ gets:

$$\mathcal{F}[F(\mathbb{I}_{2p} - \mathcal{M}_{\mathcal{G}_K})^{-1}] = \mathcal{F}\left[F \sum_{p=0}^{n} \mathcal{M}_{\mathcal{G}_K}^{\star p}[\tau]\right] = F \sum_{p=0}^{n} (\mathcal{M}_{\mathcal{G}_K}(\omega))^p \tag{C.14}$$

Provided the rank condition is satisfied, we construct the difference matrix $\Delta = H[\tau]^0 - H[\tau]^k$ and identify the source of changes in distribution and their ancestral relations. Therefore, the transitive closure $\bar{\mathcal{G}}$ of the ground truth process with its causal structure can be recovered up to the trivial transformation aforementioned. A time process without instantaneous influence must have $TC(\mathcal{G}) = \mathcal{G}$, we can recover the original process and its causal structure up to the same scaling and permutation $\pi$.

## C.3 Proof of Theorem 2

### C.3.1 Preliminary of Theorem 2

We prove our main theorem by showing that the tuple $(f, \Phi, U)$ is identifiable up to component-wise scaling and permutation. Our proof is based on the dimension of the associated variety defining special hypersurfaces in a polynomial ring $K^n$. We study the nonlinear propagation of the cumulant structure to find the identifiability conditions for INAR($\infty$) processes. Given a generic nonlinear $f$, its exact cumulant $\kappa_d(O_t)$ follows an order $d$ expansion with Bell polynomial coefficients. Accordingly, we restate Assumption 3 as follows.

**Assumption 3(restatement)** *Let observations be generated by an unknown mixing function $f$ from latent stochastic processes $Z_t^{(\Delta)}$ driven by a linear intensity $\lambda_t$. Suppose:*

1. *$f$ is a generic $C_d$ map with a full-rank Jacobian $J_f$ almost surely.*

2. *There exist at least $p$ nonzero tensors $\bar{\kappa}_d(\Delta O_t)$ for $d \in D := \{d \mid \bar{\kappa}_{d+1}(\Delta O_t) = 0\}$, where $\bar{\kappa}_d(\Delta O_t)$ is the difference cumulant computed from the first-order Taylor expansion of $O_t := f(Z_t)$.*

3. *The ideal $\mathcal{I}^\star : \langle J_f - K_{\mathcal{G}}^{(k)}(\mathbb{I}_{2p} - \mathcal{M}^{(k)}), k = 1, 2, \cdots, p \rangle$ has a zero-dimensional associated variety $\mathcal{V}(\mathcal{I}^\star)$*

**Lemma C.7.** *Let $\mathbb{Q}$ be the base field. Suppose $f$ is generic, with Jacobian matrix $J_f$. Then the following are equivalent characterizations of the genericity of $J_f$:*

1. *The entries of $J_f$ are algebraically independent commuting indeterminates over $\mathbb{Q}$; equivalently, they generate a purely transcendental extension of $\mathbb{Q}$.*

2. *The point $\big((J_f)_{i,j}\big)$ does not lie in the vanishing locus of any nonzero polynomial in the polynomial ring $\mathbb{Q}[x_{ij}]$.*

3. *Consequently, $J_f$ is of full rank on a Zariski open dense subset; in particular, it is of full rank almost surely.*

In general, as with most assumptions in identifiability analysis, the *genericity* assumption is hardly verifiable in practice. Surprisingly, it nevertheless holds *almost surely* in a probabilistic sense. To recall a simple demonstration, consider a generic matrix $F$.

When we say that a matrix is *generic*, we do not mean that it is obtained by fixing arbitrary numerical values in its entries. Instead, each entry of the matrix is regarded as a purely formal symbol—an algebraic

variable—that is not assigned any concrete value. Equivalently, we are working over the field of rational functions in these symbols, so that the entries of the matrix are algebraically independent indeterminates. This ensures that the matrix avoids all degenerate algebraic relations, except on a proper algebraic subvariety (a set of measure zero in the Euclidean sense). Thus, while the assumption is untestable numerically, it is valid almost surely under random choices, and it is rigorously formalized by treating the entries as algebraically independent symbols.

Then, we show how naturally we can assume the genericity of mixing functional, by Example 3.

*Example* 3 (Rectangular Jacobian: $f : \mathbb{R}^2 \to \mathbb{R}3$, mixed transcendental entries). Let $x = (x_1, x_2) \in \mathbb{R}^2$ and define

$$f_1(x) = \sin(\alpha_1 x_1) + e^{\beta_1 x_2},$$
$$f_2(x) = \cos(\alpha_2 x_1 + \alpha_3 x_2) + x_1^2,$$
$$f_3(x) = x_1 x_2 + e^{\beta_2 x_1 + \beta_3 x_2},$$

where all coefficients $\alpha_i, \beta_i$ are treated as algebraically independent symbols. The Jacobian $J_f(x) \in \mathbb{R}^{3 \times 2}$ is

$$J_f(x) = \begin{pmatrix} \alpha_1 \cos(\alpha_1 x_1) & \beta_1 e^{\beta_1 x_2} \\ -\alpha_2 \sin(\alpha_2 x_1 + \alpha_3 x_2) & -\alpha_3 \sin(\alpha_2 x_1 + \alpha_3 x_2) + 2x_1 \\ x_2 + \beta_2 e^{\beta_2 x_1 + \beta_3 x_2} & x_1 + \beta_3 e^{\beta_2 x_1 + \beta_3 x_2} \end{pmatrix}.$$

**Algebraic independence of the entries.** Each entry of $J_f(x)$ contains either a unique symbol factor ($\alpha_i, \beta_i$) or depends on a distinct linear combination of $x_1, x_2$. By treating the coefficients as algebraically independent symbols, one sees that no nontrivial polynomial relation among the entries can exist over $\mathbb{Q}$. Hence, the entries of $J_f(x)$ are algebraically independent over the base field $\mathbb{Q}(\alpha_1, \alpha_2, \alpha_3, \beta_1, \beta_2, \beta_3)$, so that $J_f(x)$ is generic in the sense of identifiability theory.

The generic mixing $f$ is preserved by its Jacobian matrix $J_f$; hence, identifying $J_f$ is equivalent to the recovery of $f$ up to a constant. Now, we are ready to prove our main theorem. Without loss of generality, we write $J_f$ as $F$ since they behave the same way in an algebraically closed field.

### C.3.2 PROOF OF THEOREM 2

*Proof.* For a smooth map $f : Z_t \to f(Z_t)$, we can construct $O_: = f(Z_{t+\Delta t})$ using Taylor expansion:

$$f(Z_{t+\Delta t}) = f(Z_t) + \frac{\partial}{\partial Z_t} f(Z_t) \Delta Z_t + \frac{1}{2} \frac{\partial^2}{\partial Z_t^2} f(Z_t)(\Delta Z_t)^2 + o(\Delta Z_t^3)$$

At this time, the expansion has rather abnormal behavior, as the order can be prohibitively large. However, the truncated expansion at order 1 has theoretical appeal, as we explain below. Let higher-order components be $\mathcal{R}(1)$, and recall $Z_t = H \star \epsilon_t$, we obtain the truncated differential process $\Delta \tilde{f}(Z_t)$, denoted as:

$$\Delta f(Z_t) - \mathcal{R}(1) = J_f \sum_{k=1}^{t} H_{t-s} \epsilon_s \tag{C.15}$$

We treat all quantities appearing in Eq. (C.15) as indeterminates in a polynomial ring

$$R = k\big[\{\Delta f(Z_t)\}_t, \ \mathcal{R}(1), \ J_f, \ \{H_{t-s}\}_s, \ \{\epsilon_s\}_s\big],$$

where $k$ is a base field such as $\mathbb{R}$ or $\mathbb{C}$. For each time index $t$, the defining polynomial is $g_t := \Delta f(Z_t) - \mathcal{R}(1) - J_f \sum_{s=1}^{t} H_{t-s} \epsilon_s \in R$. This polynomial generates the principal ideal $\mathcal{I}_t = \langle g_t \rangle \subset R$, and considering

all time indices $t = 1, 2, \ldots, T$, we obtain the global ideal $\mathcal{I} = \langle g_1, g_2, \ldots, g_T \rangle \subset R$. The corresponding variety is then

$$V(\mathcal{I}) = \left\{ (Z_t, \Delta f(Z_t), J_f, H_{t-s}, \epsilon_s, \mathcal{R}(1)) \in k^N \;\middle|\; g_t = 0 \text{ for all } t \right\}.$$

It is evident that $V(\mathcal{I})$ is positive-dimensional, since the defining relations do not specify finitely many points. To obtain more structure, we consider higher-order statistics. In particular, the $d$-th order cumulant tensor of the transformed increments takes the form.

$$\kappa_d\left( \Delta \tilde{f}(Z_t) \right) = \sum_{s=1}^{t} \sum_{j=1}^{p} \kappa_d^{(j)}(\epsilon) \cdot \left( J_f H_{t-s}^{(:,j)} \right)^{\otimes d}.$$

This expression shows that the cumulant naturally defines a point in the projective tensor space.

$$\mathbb{P}(V^n \otimes V^n \otimes \cdots \otimes V^n),$$

where the number of tensor factors equals $p$. Hence, while the affine variety $V(\mathcal{I})$ is too large to give identifiability, the cumulant tensors lift the problem into a projective geometric setting, where connections to secant varieties of the Veronese embedding provide a natural framework for studying uniqueness and decomposition. $\square$

Hereafter, the generic $f$ has an algebraic structure from a degenerative linear truncated cumulant. Such a degenerative form ensures that the Veronese embeddings are defined in a larger ambient space. Note that $\mathcal{R}(1)$ is completely determined by $f$ and thus can be found through an optimization problem: choosing an initial $\mathcal{R}(1)$ such that $\kappa_d\left( \Delta \tilde{f}(Z_t) \right)$ has a stable solution for a unique decomposition. We note that this optimization suggests that recovery of the entire generative model be guaranteed as long as the solution exists and is unique. One can choose any other techniques to estimate the generative model. As shown in the main text, to leverage the computational capacity of generative models, we adopt a variational method to model the causal dynamics and the mixing map.

### C.4 PROOF OF THEOREM 3

As a final remark on our identifiability theory, we prove that the proposed conditions are both sufficient and necessary for identifying the full generative model of a stochastic process. Note that when discussing a stochastic process, it has an infinite number of variables over time (*resp.*time-lag time series); therefore, we do not intend to recover the so-called causal "variables" but focus on the generative model of the process with full parameters.

$\Rightarrow$ ASSUMPTIONS LEAD TO IDENTIFIABILITY:

For sufficiency, the proof is trivial by following our proof in Theorem 2.

$\Leftarrow$ IDENTIFIABILITY INDICATES ASSUMPTIONS :

Suppose that the full generative model is identifiable even when some conditions in Assumption 3 are violated. Then there exists a linear mixing of the parameter space, $F\Phi$, that can be uniquely determined only up to a component-wise transformation $g'$ and a permutation $\pi'$ *distinct from* $\kappa(\epsilon)$ and $\pi$.

Consequently, the reconstructed observations $O_t$ satisfy

$$\kappa(O_t) = \sum_{j=1}^{p} g'(\hat{F} H_j)^{\otimes d},$$

which admits a unique decomposition.

However, this contradicts the assumed violation of the conditions in Assumption 3, because a unique decomposition should only exist when all those conditions hold. Therefore, identifiability of the full generative model implies that all the conditions in Assumption 3 must be satisfied.

## C.5 DISCUSSION FOR OTHER CASES

### C.5.1 VARYING AND DEPENDENT NOISE

We construct a model with driving noise, which is *not* time-independent but a continuous stochastic process $R_t$. To leverage the aforementioned proof, the Ito lemma is applied to the observed mixed manifold $O_t$. Plug in the causal process with the convolution kernel:

$$\Delta Z_t = (Z_{t+\Delta t} - Z_t)$$

$$= \left[ (u_0 + \int_0^{t+\Delta t} \Phi(t + \Delta t - s)Z_s \, ds + R_{t+\Delta t}) - (u_0 + \int_0^t \Phi(t - s)Z_s \, ds + R_t) \right]$$

$$= \left[ (\int_0^{t+\Delta t} \Phi(t + \Delta t - s)Z_s \, ds) - (\int_0^t \Phi(t - s)Z_s \, ds) \right] + \Delta R_t$$

For the first integral, we apply the Taylor expansion at $t - s$:

$$\Delta Z_t = Z_{t+\Delta t} - Z_t$$

$$= \left\{ \int_0^{t+\Delta t} \left[ \Phi(t-s) + \frac{\partial}{\partial \Delta t} \Phi(t-s)\Delta t + \frac{\partial^2}{\partial \Delta t^2} \Phi(t-s)(\Delta t)^2 \right] Z_s \, ds \right. \tag{C.16}$$

$$\left. - \int_0^t \Phi(t-s)Z_s \, ds \right\} + \Delta R_t$$

$$= \int_t^{t+\Delta t} \Phi(t-s)Z_s \, ds + \int_0^{t+\Delta t} \left[ \frac{\partial}{\partial \Delta t} \Phi(t-s)\Delta t + \frac{\partial^2}{\partial \Delta t^2} \Phi(t-s)(\Delta t)^2 \right] Z_s \, ds + \Delta R_t$$

$$= \Phi^* Z_t \Delta t + \Delta t \int_0^{t+\Delta t} \frac{\partial}{\partial \Delta t} \Phi(t-s)Z_s \, ds + (\Delta t)^2 \int_0^{t+\Delta t} \frac{\partial^2}{\partial \Delta t^2} \Phi(t-s)Z_s \, ds + \Delta R_t$$

$$= \Phi^* Z_t \Delta t + \Delta t (G^{(1)} \star Z_t) + (\Delta t)^2 (G^{(2)} \star Z_t) + \Delta R_t \tag{C.17}$$

**Further Remark** Without generality of a convolution, we require $s < t$, Parts for the first and second expansion are compactly written as a degraded convolution defined by a new kernel function $G^{(1)} := \Phi'(t-s)$ and $G^{(2)}$.

Then, the $(\Delta Z_t)^2$ needs an expansion of all its quadratic terms with orders of 1, 2, and 4, respectively:

$$(\Delta Z_t)^2 = \left( \Phi^* Z_t \Delta t + \Delta t (G^{(1)} \star Z_t) + (\Delta t)^2 (G^{(2)} \star Z_t) + \Delta R_t \right)^2$$

$$= \left( \Phi^{*2} Z_t^2 \Delta^2 t + \Delta^2 t (G^{(1)} \star Z_t)^2 + \Delta^4 t (G^{(2)} \star Z_t)^2 + \Delta^2 R_t \right)$$

$$+ 2 \left( AB + AC + AD + BC + BD + CD \right)$$

$$= \left( \Phi^{*2} Z_t^2 \Delta^2 t + \Delta^2 t (G^{(1)} \star Z_t)^2 + o(\Delta^4 t) + \Delta^2 R_t \right) \tag{C.18}$$

$$+ 2 \left( AB + o(\Delta^3 t) + AD + o(\Delta^3 t) + BD + CD \right) \tag{C.19}$$

We shall argue that all terms in $(\Delta Z_t)^2$ are obtained via Taylor expansion on the kernel $\Phi(t-s)$ and $f(Z_{t+\Delta t})$. Therefore, derivation of the Ito Lemma from convolution kernels needs meticulous study of the order of each infinitesimal and their limiting distributional behavior with respect to the order of increments in random noise. Next, we show that the order of incremental perturbation is tightly coupled with the degree and difficulty to which the full identifiability of the latent causal stochastic process can be achieved. Since the noise is assumed white, that eliminates all dependence over time but still allows for heterogeneity.

### C.5.2 IDENTIFICATION WITH INSTANTANEOUS INFLUENCE

In this section, we give a brief discussion on cases in which the model has instantaneous influences, and we highlight the complexity of recovering the entire generative model, which agrees with the complexity of solving a system of quadratics.

Identifiability results have been shown in our main theorem, given a model without instantaneous influence. In such a case, $\mathcal{M}$ is not only an upper-triangular matrix whose entries as indeterminates in a projective variety $\mathbb{P}^{4p^2-2p-1}$ defined by a characteristic polynomial, but indeed a matrix with strictly non-zero entries in the upper-right block of size $p \times p$. Consider the augmented matrix $F_{\mathscr{G}}$ in the form of:

$$
\begin{bmatrix}
\vdots & \vdots & \vdots & \vdots & \vdots & \vdots \\
f_1 & \cdots & f_q & e_{q+1} & \cdots & e_{2q} \\
\vdots & \vdots & \vdots & \vdots & \vdots & \vdots \\
\hline
\vdots & \vdots & \vdots & \vdots & \vdots & \vdots \\
e_1 & \cdots & e_q & f_1 & \cdots & f_q \\
\vdots & \vdots & \vdots & \vdots & \vdots & \vdots
\end{bmatrix}
$$

We regard this augmentation matrix as generic because, in the projective space $\mathbb{P}^d$, almost every point corresponds to a configuration in which no two directions collapse, i.e., the associated lines intersect only at infinity and thus do not exhibit linear degeneracy. The augmentation matrix is generic in the sense of lying in a Zariski-open dense set of $\mathbb{P}^d$, so replacing $F_{\mathscr{G}}$ with its augmented form does not alter $\dim(V)$. However, in the presence of instantaneous influences, the defining equations impose algebraic constraints that collapse this open set. In this case, the only admissible augmentation corresponds to forcing all other blocks to vanish, hence no non-trivial generic matrix can be constructed. Assume the INAR model with instantaneous influences encoded in a matrix $B$. Such a model is presented as follows,

$$
BX_t = A_1 X_{t-1} + A_2 X_{t-2} + \cdots + A_p X_{t-p} + u_t, \tag{C.20}
$$

where:

$\qquad B$ is a non-diagonal matrix capturing contemporaneous (instantaneous) effects among variables.

$\qquad u_t$ represents structural shocks, often assumed to satisfy $\mathrm{Cov}(u_t) = I$.

It follows that the infinite order model is

$$
O_t = FB^{-1}(\mathbb{I}_p - \Phi)^{-1} \star \epsilon_t \tag{C.21}
$$

It is evident that, following our reasoning, one can still recover the mixed parameter space $K' = FB^{-1}(\mathbb{I}_p - \Phi)^{-1}$ up to scaling and permutation. However, the obtained matrices constitute a degree 3 polynomial system that needs more constraints to cut out the individual parameter spaces.

**Claim 1.** *For the INAR model with instantaneous influences, Theorem 2 is never sufficient to recover the entire generative model.*

# D  EXTENDED IDENTIFIABILITY RESULTS

## D.1  EXISTENCE OF HIERARCHY MINIMALITY

We have shown that causal disentanglement under INAR ($\infty$) is guaranteed by Theorem 1 and Theorem 2. This problem is then reduced to finding the minimal cumulant hierarchical structure (complexity) and searching $d$ to minimally achieve this complexity. As an inspiration, we also show that the algebraic geometry properties uniquely determine the minimality of cumulant complexity. Therefore, the Identifiability of latent causal structure can be controlled from the geometric perspective. Then, we present several demonstrations to find the minimal complexity under any data manifolds.

*Remark* 1. In this section, we tentatively ignore our weak convergence class and consider all generalized situations where weak convergence is no longer required. As a simple demonstration, we consider linear and full-rank polynomial mixing of a latent causal time process driven by any noise family as a variation of Theorem 1.

We first discuss a spectrum of variations for different noise processes as a path-wise nondifferentiable process or a semi-martingale. We recall two basic identification theories and elucidate our theorem spans them strictly by finding different hierarchy minimality.

**Proposition 2** (further identifiability under special noise ). *Given a sequence of stochastic processes defined in Theorem 2, the latent causal representation $\mathcal{G}$ is identifiable up to the sign flipping and a permutation if and only if the noise follows a Laplacian distribution.*

**Proposition 3** (reduced identifiability under linear mixing $F$, adapted from Carreno et al. (2024)). *Given a sequence of stochastic processes defined in Theorem 2, the latent causal structure $\mathcal{G}$ is identifiable up to scaling and permutation.*

Now, we focus on a more complicated but more useful scenario in which the noise process arbitrarily behaves.

**Theorem A4** (identification on the limited noise support).

> *(1.) Under Theorem 1, the identifiability is guaranteed by finding $d_0$ to satisfy the minimal cumulant hierarchy proposed in Assumption 2.*
>
> *(2.) $d_0$ and $T$ is tractable.*
>
> *(3.) Assumption 2 is still both sufficient and necessary.*

According to standard tensor algebra, a linear transformation $F$ propagates cumulant information from the observed space to the latent space either via a multilinear transformation—when $F$ is a linear map—or via a multi-polynomial mapping—when $F$ is a full-rank polynomial function. In the latter case, the resulting system of polynomial equations grows combinatorially in complexity, reflecting the interaction between the polynomial structure and the higher-order cumulants of the latent variables. Notably, all such equations are governed by the same scale and distributional structure of the underlying noise process.

## D.2  PROOF OF PROPOSITION 2

We adopt a **frequency-domain transformation** to characterize both continuous-time causal influences and standard causal transitions. Importantly, this transformation is applied *only in the latent space*, while the observables remain real-valued in the time domain, consistent with real-world data.

***Remark*** *In the frequency domain, the variable $f$ serves as a continuous index that characterizes the spectral behavior of a signal. However, distributional shifts in this domain—especially those involving*

*complex-valued structures—often manifest at fine-grained levels that standard likelihood-based density modeling fails to capture. Instead, statistical representations such as the **power spectral density** (PSD) and its higher-order extensions (e.g., bispectrum, trispectrum) provide more faithful characterizations of the distribution's structural dependencies across frequencies. These quantities are directly connected to the underlying cumulants of the signal and thus offer a natural multiscale lens to detect and interpret intervention-induced shifts. Unlike traditional criteria, such as sufficient variability, which are often coarse, higher-order cumulants and spectra enable more granular identification of structural changes at each statistical order.*

Let $S_X(f)$ denote the power spectral density of a random time-domain signal $X_t$, and define the autocorrelation function as

$$R_X(\Delta) := \mathbb{E}[X_t X_{t+\Delta}].$$

According to the Wiener–Khinchin theorem, $S_X(f)$ is the Fourier transform of $R_X(\Delta)$.

We apply this PSD analysis to the transformed latent variable defined as a convolution:

$$Z_t^\Delta = K_t \star R_t,$$

and analyze its structure in the frequency domain.

$$S_Z(f) = S_{K \star R}(f) = ||K(f)||_F^2 S_R(f) \tag{D.1}$$

$$= ||K(f)||_F^2 S_{\epsilon+u}(f)$$

$$= ||K(f)||_F^2 (S_\epsilon(f) + S_u(f))$$

$$= ||K(f)||_F^2 \frac{1}{2\pi} \int_G R_R(\Delta) e^{-i\Delta f} \, d\Delta \tag{D.2}$$

The first equation features the relation between the origin and the composite signal to which a filter kernel is applied. One followed by the next two equations obtained by applying the Fourier transformation to the field $G$ that is separated into two cases: $\Delta = 0$ and otherwise. The result above is called the Power Spectrum Density matrix, each entry $[S_Z(f)]_{ij}$ describing the entire density information for all time lags $\Delta$. $\delta(f)$ is a Dirac delta function and $A^H$ is the Hermitian transpose. Our identifiability is built upon the reasoning in the latent space PSD and the reconstruction equivalence between $Z_t^{(\Delta)}$ and $\hat{Z}_t^{(\Delta)}$.

We start with an overly simplified case indicated by Eq.(D.5). Note that the power spectrum density matrix serves as a representation of the distribution of the total power in different frequencies, and thus it has cross-spectrum components that capture interactions among different latent stochastic processes. Let $S_Z(f)$ be a diagonal matrix such that all cross-spectra in the frequency domain disappear, such that

$$[S_Z]_{ij} = \sum_{j=1}^p \sum_{i=1}^p K_{ik} S_{R,kl} \overline{K_{lj}} = 0, \ i \neq j \tag{D.3}$$

We begin our technical analysis by revisiting the change-of-variable transformation, a fundamental technique frequently employed in causal representation learning frameworks. Here, we derive its counterpart in the frequency domain.

**Fact 2.** Let $Z_t$ be a real-valued latent process and let $h := \hat{f} \circ f^{-1}$ be the reconstruction map applied to $Z_t$. If the noise is Laplacian-distributed, then the power spectral density (PSD) of the transformed differential process $dh^{-1}(Z_t) := d\hat{Z}_t$ satisfies:

$$S_{dh(Z)}(f) = J_{h^{-1}} S_{dZ}(f) J_{h^{-1}}^T, \tag{D.4}$$

where $J_h$ is the Jacobian of $h$.

*Proof.* From the definition of the power spectral density, we have:

$$S_Z(f) = \frac{1}{2\pi} \int_0^\infty R_Z(\Delta) e^{-i\Delta f}\, d\Delta,$$

where $R_Z(\Delta) := \mathbb{E}[Z_t Z_{t+\Delta}^T]$ denotes the autocorrelation function. For the transformed process $h(Z_t)$, its autocorrelation is:

$$R_{h(Z)}(\tau) = \mathbb{E}[h(Z_t) h(Z_{t+\tau})^T].$$

Applying the change-of-variable formula to the differential $dh(Z_t)$, we need:

$$\mathbb{E}[dh(Z_t) dh(Z_{t+\tau})^T] = J_h\, \mathbb{E}[dZ_t dZ_{t+\tau}^T]\, J_h^T.$$

That holds when the map $h$ is an affine, meaning $h^{-1}$ must be a linear map, a result adapted from Klindt et al. (2021). Therefore, the autocorrelation function of the transformed variable is linearly related to that of $Z_t$ through the Jacobian, and so is the power spectral density via the Fourier transform. This proves the identity in Eq. (D.4). □

By the derived PSD equivalence formula, a similar condition as in a regular time-domain probability space is satisfied if a learner encoder matches the distribution between the estimated and ground variable:

$$S_{\hat{Z}:=h(Z)}(f) = S_Z(f) \tag{D.5}$$

Plug in all terms we have derived beforehand:

$$\mathcal{F}\{(I - \Phi_t)^{-1}\}(S_\epsilon + uu^T \delta(f))\mathcal{F}\{(I - \Phi_t)^{-1}\}^H = J_h S_Z J_h^T \tag{D.6}$$

Getting each entry of this relation:

$$\begin{bmatrix} K_{f,11} & \cdots & \cdots \\ \vdots & \ddots & \vdots \\ K_{f,1p} & \cdots & K_{f,pp} \end{bmatrix} \begin{bmatrix} S_\epsilon(f)_{11} + u_1^2 \delta(f) & \cdots & \cdots \\ \vdots & \ddots & \vdots \\ S_\epsilon(f)_{p1} + u_p u_1 \delta(f) & \cdots & S_\epsilon(f)_{pp} + u_p u_p \delta(f) \end{bmatrix} \begin{bmatrix} \overline{K_{f,11}} & \cdots & \cdots \\ \vdots & \ddots & \vdots \\ \overline{K_{f,1p}} & \cdots & \overline{K_{f,pp}} \end{bmatrix}^T \tag{D.7}$$

$$= J_h S_Z J_h^T \tag{D.8}$$

where the $\overline{X(f)}$ is the complex conjugate of the spectrum with respect to $f$ and $\phi_{ij}$ the filter kernel function. Next, we focus on the identifiability of the ground true stochastic process $Z_t$.

If the immigrant parameter $u = 0$ and the noise processes are white, that means their variance is not time-varying. Then $S_R$ is a diagonal matrix in that Eq.(D.2) has a more explicit form:

$$||K||_F^2 \frac{1}{2\pi} \int_0^\infty \mathbb{E}[R_t R_{t+\Delta}] e^{-i\Delta f}\, d\Delta$$

$$= ||K||_F^2 \frac{1}{2\pi} \int_G \mathbb{E}[(\epsilon_t)\epsilon_{t+\Delta})]\delta(\Delta) e^{-i\Delta f}\, d\Delta \tag{D.9}$$

$$= ||K||_F^2 \frac{1}{2\pi}\{\int_{G=0} \mathbb{E}[(R_t)(R_{t+\Delta})]\delta(t) e^{-i\Delta f}\, d\Delta + \int_{G\neq 0} \mathbb{E}[(R_t)(R_{t+\Delta})]\delta(t) e^{-i\Delta f}\, d\Delta\} \tag{D.10}$$

$$= \mathcal{F}\{(I - \Phi_t)^{-1}\}\Sigma_{\epsilon_t}\mathcal{F}\{(I - \Phi_t)^{-1}\}^H \tag{D.11}$$

Therefore, to make the condition hold, the only solution is to make $K$ (consequently $K^H$) a diagonal matrix, causing $K$ and $K^T$ should be a permutation scaling matrix. Namely, we have:

$$P_\pi \begin{bmatrix} K_{f,11} & \cdots & \cdots \\ \vdots & \ddots & \vdots \\ 0 & \cdots & K_{f,pp} \end{bmatrix} \Sigma \begin{bmatrix} K_{f,11} & \cdots & \cdots \\ \vdots & \ddots & \vdots \\ 0 & \cdots & K_{f,pp} \end{bmatrix}^H P_\sigma = J_h S_Z J_h^T \tag{D.12}$$

The inverse of $K$ is still a permutation scaling matrix, so it means $\mathcal{F}\{(I - \Phi_t)^{-1}\}$ also has only one non-zero value in each column and row. A stochastic delayed process should at least be correlated to itself, so $[(I - \Phi_t)_f^{-1}]_{ii} \neq 0$ and make $I - \Phi_f$ a permutation scaling matrix only when its $(i, j)$ entry is 0. Therefore, $I - \Phi_f$ and its inverse $K$ are both diagonal and

$$\phi_{f,ij} = 0, i \neq j \tag{D.13}$$

which is equivalent to $\int_G \phi(t)_{ij} e^{2\pi it} dt = 0$ and means $\phi(t)_{ij} = 0$. By construction, the RHS should thus be diagonal, and so is $S_{\hat{Z}}$. It reduces the condition to:

$$\Lambda_Z = J_h \Lambda_Z J_h^T \tag{D.14}$$

We can always left multiply $J_h^{-1}$ and right multiply $J_h^{-T}$ to get $J_h^{-1} \Lambda_Z J_h^{-T} = \Lambda_Z$. Since $\Lambda_Z$ is a diagonal matrix, then $J_h$ must not have more than one non-zero element, indicating $J_h$ can be written as $P_\sigma \mathrm{diag}(k_1, k_2, k_3, \ldots, k_p)$ and thus a mixing of permutation and a scaling matrix. We now show that the identifiability can be further improved. Since we have established that $K_f S_R K_f^H = J_h K_f S_R K_f^H J_h^T$ and fixed both sides to be diagonal. If we first consider a real permutation scaling matrix of order 2, then we have:

$$P_\pi D(K_{11}, K_{22}, K_{33}) \Lambda(\sigma_{11}, \sigma_{22}, \sigma_{33}) D(K_{11}, K_{22}, K_{33}) P_\sigma = Q \Sigma Q^T \tag{D.15}$$

Since $P_\sigma = P_\pi^T$, we also get $PD\Lambda DP^T = (PD)\Lambda(DP^T) = (DP)\Lambda(P^T D) = D(P\Lambda P^T)D$. The resulting matrix is a similarity transformation that scales by the value of $K_{ii}$ each entry in the original noise spectrum density matrix. This relationship still holds when considering a complex-valued spectrum density matrix as in our setting, because what only needs to be changed is to replace $a$ with $a + bi$, resulting in each entry of $\Lambda$ scaled by $(a + pi)\overline{(a + pi)} = a^2 + p^2 = A, (b + mi)\overline{(b + mi)} = b^2 + m^2 = B$, and similarly $C$:

$$\begin{bmatrix} a + pi & 0 & 0 \\ 0 & 0 & c + qi \\ 0 & b + mi & 0 \end{bmatrix} \begin{bmatrix} \sigma_{11}^2 & 0 & 0 \\ 0 & \sigma_{22}^2 & 0 \\ 0 & 0 & \sigma_{33}^2 \end{bmatrix} \begin{bmatrix} a - pi & 0 & 0 \\ 0 & 0 & b - mi \\ 0 & c - qi & 0 \end{bmatrix} \tag{D.16}$$

$$= \begin{bmatrix} A\sigma_{11}^2 & 0 & 0 \\ 0 & C\sigma_{33}^2 & 0 \\ 0 & 0 & B\sigma_{22}^2 \end{bmatrix} \tag{D.17}$$

We know the fact that $J_h$ is a permutation scaling matrix such that $\Lambda = J_h \Lambda J_h^T$. This means the Jacobian cannot change the values in the main diagonal. Leveraging a permutation scaling matrix magnifies the main diagonal entry and reorders the elements. Let the entry of the permutation scaling Jacob be $A, B$, and $C$. We can conclude $A^2 = B^2 = C^2 = 1$. This leads to the component-wise identifiability up to a permutation and sign flipping. The permutation is due to the random labeling of each process, for example, letting $K_{11}$ to $K_{22}$, which does not change the diagonal form of the matrix but just re-numbers the process.

### D.3 PROOF OF PROPOSITION 3

As one of the most interesting results for identifiability under a generic map $f$, we show our theorem covers the autoregressive model as a special case but requires less strict conditions on $f$. We adapt notations from (Yao et al., 2022a) and recall some important notations for $f, g$ and latent variable $z_t = \{z_t^i\}_{i \in p}$:

$$x_t = g(z_t); z_t^i = f_i(\{z_{j,t-\tau} | z_{j,t-\tau} \in \mathbf{Pa}(z_{it})\}, \epsilon_t^i, \theta_r^c, \theta_r^o) \tag{D.18}$$

where $f$ is a generic map and $g = \sum_{\tau=1}^{p} B_\tau Z_{t-\tau} + E_t$ is an autoregressive causal model of order $\tau$ (i.e., conditional independence holds for every $t - \tau$ time stamps). Since it is a VAR ($p$) model, no convergence is needed for identification of the latent model.

To connect our work to a sufficient number of prior works on causal representation learning and time-delayed causal models, we present an analogue, time-varying time series and filter systems to standard autoregressive time-delayed causal models. We view a continuous-time series $X_t$ as a source signal passed to a filter $A(L)$, which is a real-valued matrix or matrix-value functions, to generate a new signal series:

$$Y_t = A(L)X_t + B_t \tag{D.19}$$

that can be expanded more explicitly as $Y_t = \sum_{k=1}^{T} A_k X_{t-k} + B_t$. $A_k$ is a matrix whose values may vary with time, and $B_t$ is another random, noisy process. In many applications, $B_t$ represents a Brownian motion or a stationary $I(0)$ Poisson process. These processes are widely considered in stochastic differential equation systems that reflect the random motion of a physical object, such as particles or molecules. One more special case is to choose $B_t := \text{Bernoulli}(p, q)$ which changes $Y_t$ into an integer-value autoregressive process (INAR) accompanied with a thinning operator $A \circ Y$ (Weiß, 2008). No dependence between the filter matrix and time leads to a reduced form of autoregressive models.

Our theory supports VAR models with any fixed time lag by identifying the VAR($\infty$) first and letting $\tau_{\geq p} = 0$. Since the second step of our proof induces a GL representation and thus 0 will always be mapped to the original 0 in the kernel, completing the proof.

# E   DETAILED MUTATE CONFIGURATION

**Comparison to methods of learning latent causal variables**   Throughout this paper, the identifiability guarantees hold for any generic map $f$. In particular, our framework allows multiple latent processes $Z_t^{i,j,k}$ to be mapped to a smaller number of observed variables $O_t^{s,m}$, as well as a single latent process $Z_t^i$ to be mapped to multiple observations $O_t$. We intend to compare our model to those focusing on the recovery of latent causal variables. Therefore, we also use an invertible mixing in simulations.

## E.1   SIMULATION REGIME

We demonstrate the generative process for INAR equivalent classes. For a fair comparison to those baselines mainly addressing step-wise conditional independence, we generate for both time-step dynamics and denser dynamics by changing the setup to very short kernel effects with $\tau \in (0.001, 0.01) = t - t'$. We generate stochastic point processes from three basic kernel response functions:

$$\phi_{\text{exponential}}(t) = \alpha e^{-\beta t'}, \alpha \sim \text{uniform}(0.1, 0.5) \text{ and } \beta \sim \text{uniform}[0.5, 2]$$

$$\phi_{\text{powerlaw}}(t) = \frac{\alpha}{(t+c)^\beta} \cdot \mathbf{1}, \alpha \sim \text{uniform}(0.5, 1.2), \beta \sim \text{uniform}[0.1, 0.8] \text{ and } \gamma \in \text{uniform}(1, 3, 1.8)$$

$$\phi_{\text{rectangular}}(t) = \frac{1}{T - T'} \cdot \mathbf{1}_{\{t' \leq T\}}$$

The baseline intensity $u_0$ is sampled from $\text{uniform}(0, 1, 0.2)$. All parameters of the basic kernel are uniformly sampled by ensuring $\alpha < \beta$ with exponential response, $\alpha < \gamma$ with a power-law response, respectively, to satisfy the stationary increment condition such that $|\phi| < 1$. In simulations, we also consider two extreme cases for simple nonlinear intensity and nonparametric intensity. We construct the conditional intensity

function by mixing latent features through a linear transformation followed by a non-linear activation. Specifically, we first compute a log-linear intensity using the expression

$$\lambda_t = \log(1 + \exp(z_\ell - r_\ell[:, \Delta, :]))$$

that ensures positivity and controls the scale of the output through a smoothed ReLU (i.e., softplus).In an alternative setting (`kernel == "np"`), we learn the intensity function using a small neural network (MLP): a two-layer perceptron with ReLU activation, ending in a Softplus to maintain positive outputs. This setup enables flexible, data-driven modeling of intensity dynamics beyond purely additive or linear forms. We define the mixing intensity function using a two-layer feedforward neural network with ReLU and Softplus activations. Formally, the architecture is given by:

$$\lambda_t = \sigma_+ \left( W_2 \cdot \text{ReLU}(W_1 \lambda_t(l) + b_1) + b_2 \right), \tag{E.1}$$

where

- $\lambda_t(l) \in \mathbb{R}^d$ is the input linear basic intensity at time $t$,
- $W_1 \in \mathbb{R}^{64 \times d}$, $b_1 \in \mathbb{R}^{64}$ are the weights and bias of the first layer,
- $W_2 \in \mathbb{R}^{d \times 64}$, $b_2 \in \mathbb{R}^d$ are the weights and bias of the second layer,
- $\sigma_+(x) := \log(1 + e^x)$ denotes the Soft-plus activation.

This design ensures the output $\lambda_t$ remains strictly positive and can model complex dependencies in the latent dynamics while maintaining numerical stability.

We model the transformation from the latent variable $Z_t \in \mathbb{R}^d$ to the observational space via a multi-layer mixing network. Specifically, for each layer $l = 1, \ldots, L-1$, the transformation is given by $Z_t^{(l)} = \mathbf{A}^{(l)} \cdot \sigma_{\text{leaky}}(Z_t^{(l-1)})$, where $\mathbf{A}^{(l)} \in \mathbb{R}^{d \times d}$ is an orthogonal mixing matrix and $\sigma_{\text{leaky}}$ denotes the leaky ReLU activation with slope $\alpha = 0.2$. The initial input is $Z_t^{(0)} = Z_t$, and the final output $Z_t^{(L-1)}$ represents the observation-space signal.

### E.2  PRIOR DECOMPOSITION OF TIME-ADAPTIVE MODULE

Without loss of generality, we consider non-finite steps for a latent stochastic generative process, as discussed in Lemma 2, where $\Delta t \to 0$. This induces an equivalence that the intrinsic history—the filtration $\mathcal{F}_t := \sigma \left( \bigcup_{0 < t < T} \sigma(Z_t^\Delta) \right)$—ensures that the process $Z_t^{(\Delta)}$ is $\mathcal{F}_t$-adaptive and measurable.

We decompose the ELBO objective as follows:

$$\text{ELBO} = \log p(O) - D_{\text{KL}}(q_\phi(Z|O) \| p(Z))$$

$$= \mathbb{E}_{z \sim q(Z_t|O_t)}[\log p(O_t|Z_t)] + \mathbb{E}_{z \sim q(Z_t|O_t)} \left[ \log \frac{q(Z_t|O_t)}{p(Z_t)} \right]$$

$$= \mathbb{E}_{z \sim q(Z_t|O_t)}[\log p(O_t|Z_t)] - \mathbb{E}_{z \sim q(Z_t|O_t)} \left[ \log q(Z_t|O_t) - \log p(Z_t) \right]$$

$$= \mathbb{E}_{z \sim q(Z_t|O_t)} \left[ \log p(O_t|Z_t) - \log q(Z_t|O_t) \right] + \mathbb{E}_{z \sim q(Z_t|O_t)} \left[ \log p(Z_t) \right]$$

$$= \mathbb{E}_{z \sim q(Z_t|O_t)} \left[ \log p(O_t|Z_t) - \log q(Z_t|O_t) \right] + \mathbb{E}_{z \sim q(Z_t|O_t)} \left[ \sum_{\mathcal{F}_0^+}^{\mathcal{F}_T} \log p(Z_t^{(\Delta)}|\mathcal{F}_t) \right]$$

The reason we can segment the increasing filtration in the last term is due to the nice property of $\mathcal{F}_t$-measurable sequence. We can show the filtration of $Z_t|Z_s, R_t$ and $Z_t|R_s$ is equal because it is well known that any

$p$-order INAR sequence with stationary increments admits a moving average (MA) representation. The further construction of their filtration $\tilde{\mathcal{F}}_t(\text{resp.}R_{s<t})$ and $\mathcal{F}_t(\text{resp.}Z_{s<t}, R_t)$ can show

$$\tilde{\mathcal{F}}_t = \mathcal{F}_t$$

We prove the result in the sequel. For $\tilde{\mathcal{F}}_t$, $Z_t$ is a measurable function for $s < t$. By causality of the convolution kernel $\Psi = (I - \Phi)^{-1}$ satisfying $\Psi_\tau = 0$ for $\tau < 0$, which indicates $Z_t \in \sigma(R_s : s < t)$. Then, we construct another filtration $\tilde{\mathcal{F}}_s : \sigma(R_u : u < s)$. By adaptivity, $\tilde{\mathcal{F}}_s : \sigma(R_u : u < s) \subseteq \tilde{\mathcal{F}}_t : \sigma(R_u : u < t)$. Therefore, $Z_s$ is also $\sigma(R_u : u < t)$-measurable. Since the minimal $\sigma$-algebra of the original $\mathcal{F}_t$ measurable function must be contained in its $\sigma$-algebra, we have $\sigma(Z_s) \subseteq \sigma(R_u : u < t)$ and $\sigma(\bigcup_{s \leq t} \sigma(Z_s)) \subseteq \sigma(R_u : u < t)$. For $\mathcal{F}_t$, $R_t = Z_t - \Psi \star Z_t$ so $R_t$ is $\sigma(Z_s : s \leq t)$ measurable. Therefore, by a similar construction, it is evident that $\sigma(\bigcup_{s<t} \sigma(R_s)) \subseteq \sigma(Z_s : s \leq t)$. Therefore, because $\tilde{\mathcal{F}}_t \subseteq \mathcal{F}_t$ and $\mathcal{F}_t \subseteq \tilde{\mathcal{F}}_t$, there must be $\tilde{\mathcal{F}}_t = \mathcal{F}_t$.

Following this set-up, the prior becomes:

$$Z_t \mid \mathcal{F}_t \sim \mathcal{N}\left(\begin{bmatrix} u_1(t) \\ u_2(t) \\ \vdots \\ u_p(t) \end{bmatrix} \sum_{t'<t}(I - \Phi)^{-1}, \ \sum_{t'<t}(I - \Phi)^{-1}\Sigma_{t'}(I - \Phi)^{-T}\right)$$

The latents are generated by $Z_t = (I - \Phi) \star R_t$, where $R_t$ is modeled as isotropic Gaussian noise with mean $U$ and variance $\Sigma$. Note that the variance matrix $\Sigma_{Z_t}$ is zero for any $t - t' \neq 0$. By Wiener-Khinchin Theorem (Wiener, 1949), we have the covariance matrix $C_{Z_t}(0) = \frac{1}{N}\sum_{k=0}^{N-1} S_z(w_k)$, we drop the sub-index $Z_t$ whenever the context is clear. Now we can derive the decomposition of the convolution prior as

$$\mathbb{E}_{z \sim q(Z_t|O_t)}\left\{\sum_{\mathcal{F}_0^+, Z_t}^{\mathcal{F}_T} \log p(Z_t^{(\Delta)}|\mathcal{F}_t)\right\}$$

$$= \mathbb{E}_{z \sim q(Z_t|O_t)}\left\{\sum_{\mathcal{F}_0^+, Z_t}^{\mathcal{F}_T} \log p\left[(I - \Phi_t) \star \hat{R}_t^{(\Delta)}\right]\right\} = \mathbb{E}_{z \sim q(Z_t|O_t)}\left\{\sum_{\mathcal{F}_0^+, Z_t}^{\mathcal{F}_T} \log p\left[\int_0^t (I - \Phi_{t-t'})\hat{R}_{t'}^{(\Delta)}\, dt\right]\right\}$$

$$= \mathbb{E}_{z \sim q(Z_t|O_t)}\left\{\sum_{\mathcal{F}_0^+, Z_t}^{\mathcal{F}_T} \log p\left[\mathcal{N}(\hat{U}_R, \sum H_t^{-1}\Sigma_{\hat{R}_t'}H_t^{-T})\right]\right\}$$

$$= \mathbb{E}_{z \sim q(Z_t|O_t)}\left\{\sum_{\mathcal{F}_0^+, Z_t}^{\mathcal{F}_T} \log p\left[\mathcal{N}(\hat{U}_R, \underbrace{\sum H_t^{-1}(PSD_{\hat{Z}_t})\Sigma_{\hat{R}_t'}H_t^{-T}(PSD_{\hat{Z}_t})}_{C_{p(Z_t)}(0)})\right]\right\}$$

$$= \mathbb{E}_{z \sim q(Z_t|O_t)}\left\{\sum_{\mathcal{F}_0^+, Z_t}^{\mathcal{F}_T} \log p\left[\mathcal{N}(\hat{U}\sum_{t'<t}\underbrace{(I - \Phi(t - t'))^{-1}}_{(1-\Phi)(w_k)^{-1}\Sigma(1-\Phi)(w_k)^{-H}=S_Z(w_k)}, \ \frac{1}{N}\sum_{k=0}^{N-1} S_{Z_t}(w_k))\right]\right\} \quad \text{(E.2)}$$

$$= \mathbb{E}_{z \sim q(Z_t|O_t)} \left\{ \sum_{\mathcal{F}_0^+, Z_t}^{\mathcal{F}_T} \log p \left[ \mathcal{N}(\hat{U} \underbrace{\sum_{\tau>0,w=0} (I - \Phi(\tau))^{-1} e^{-jw\tau}}_{\text{the inverse Fourier at } w=0}, \frac{1}{N} \sum_{k=0}^{N-1} S_{Z_t}(w_k)) \right] \right\}$$

$$= \mathbb{E}_{z \sim q(Z_t|O_t)} \left\{ \sum_{\mathcal{F}_0^+, Z_t, N \in (N_0, T)}^{\mathcal{F}_T} \log p \left[ \mathcal{N}(\hat{U} \mathrm{PSD}_{Z_t}(H(0)), \frac{1}{N} \sum_{k=0}^{N-1} S_{Z_t}(w_k)) \right] \right\} \tag{E.3}$$

## E.3 EXPLICIT CONTROL FOR CONVOLUTION PRIOR

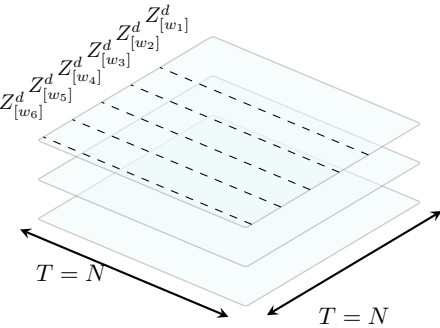

Figure A4: Visually Time-adaptive PSD Computation

**Remark.** The summation of kernel products and integrated noise variables is guaranteed to converge to the true time-adaptive process under $\mathcal{F}_t$, provided that the time discretization is sufficiently dense. The latent variable $Z_t^\Delta$ is sampled from the encoder distribution $q_\phi$ and passed to the PSD decomposition module to compute the frequency-domain representation of the full kernel matrix $F_w[1 - \Phi_t]$ and the power spectral density $S_{\hat{R}_t}$.

**Encoder-PSD flow** As shown in Eq. (E.3), a key component of our module is to efficiently compute the decomposition of the PSD matrix. However, under milder regularity conditions, the PSD decomposition is not unique, and thus can only be recovered up to the minimal-phase. Therefore, the encoded distribution is not sufficient to decompose the PSD matrix for which a reparameterization is needed. An encoder receives a $T$-length sequence $O_t$ and returns the latent variable vector. Fast Fourier Transformation converts the latent sequence to a vector of equal length up to $t$:

$$[Z_{\mathcal{F}_0}, Z_{\mathcal{F}_t}, \cdots, Z_T] \Rightarrow \{[Z[f_0], Z[f_k], \cdots, Z[K]] | K = 0, 1, 2, \cdots, T\},$$

and the flow method is enforced by solving the following Wilson Factorization optimization problem for each $[Z[f_0], Z[f_k], \cdots, Z[K]]$, finding the transferring matrix

$$H^\dagger = \arg \min_{\Sigma_t = \sigma^2 I} PSD(Z_t) - H^\dagger \Sigma_t H^{\dagger H}.$$

It is then sent to evaluate the true prior distribution, supporting the joint optimization of all loss components. The prior distribution is more complicated when a PSD decomposition program is used. We evaluate the

causal prior with a discretized filtration, which otherwise should be defined path-wise. The continuous filtration (or the intrinsic history) $\mathcal{F}_t$ is estimated as $\mathcal{F}_{t_-}^{\Delta}$ controlling the subsequence operator $\Delta$. This module shares conceptual similarities with prior causal representation learning (CRL) approaches that rely on posterior inference. However, a key distinction lies in the fact that we do not require an explicit transition map of the form $f : [Z_{HX}, \epsilon_t] \rightarrow [Z_{HX}, Z_t]$. Instead, due to the self-convolution structure and the convergence guarantees we establish, the transition is implicitly realized without additional modeling overhead. Crucially, the causal structure within the latent space is preserved in an explicit form.

We further remark that the key step, spectrum decomposition, is completed for the entire encoded trajectory $\hat{Z}_{t_0:T}$, and the prior structure is ensured by segmenting filtration. This features the major difference in prior work that recursively constructs an equal-length sliding window for each latent. Filtration segmentation can work with causal masks that a more expressive encoder leverages. Note that transformer modules are not a required component for shorter sequences, i.e. $T < 100$. However, when the sequence is extremely long as simulated in the conventional class of stochastic point processes, a transformer can be used in place of a common MLP encoder to learn much more expressive latent embeddings by utilizing the filtration attention from arbitrarily long past events.

**Overall training loss**    To encourage sparsity in transferring kernels, we follow the widely-used penalty to jointly optimize:

$$\mathcal{L}_{Total} = \mathcal{L}_{\text{Recon}} - \beta \mathcal{L}_{\text{KLD}} - \gamma |\Phi| - \omega \mathcal{L}_{\text{PSD}} \tag{E.4}$$

This training objective ensures the learned latent process is driven by a family of generalized white processes, as, in the Encoder-PSD flow, the decomposition is enforced by the prescribed isotropic noise, which omits any discriminator module as used in Yao et al. (2022b). The coefficients in sparsity loss and PSD accuracy are registered as tunable hyperparameters.

### E.4    EXTENDED RESULTS

Table A2: Reporting best performance for each baseline

| Method | Metric | Kernel Ave. | Exp | Power. | Rect. | Nonlin. | Nonpar. |
|---|---|---|---|---|---|---|---|
| TDRL | MCC | 0.657 | 0.629 | 0.653 | **0.773** | 0.584 | **0.644** |
| | $\mathcal{L}_{vae}$ | **0.449** | **0.308** | **0.302** | 0.302 | 0.871 | **0.461** |
| BetaVAE | MCC | 0.419 | 0.395 | 0.414 | 0.420 | 0.433 | 0.433 |
| | $\mathcal{L}_{vae}$ | 9.480 | 8.538 | 7.533 | 8.424 | 11.683 | 11.220 |
| SlowVAE | MCC | 0.410 | 0.384 | 0.405 | 0.420 | 0.425 | 0.412 |
| | $\mathcal{L}_{vae}$ | 362.890 | 395.107 | 448.105 | 452.472 | 238.520 | 280.247 |
| PCL | MCC | 0.440 | 0.469 | 0.379 | 0.430 | **0.474** | 0.449 |
| | $\mathcal{L}_{vae}$(train) | 0.693 | 0.693 | 0.694 | 0.693 | **0.693** | 0.693 |
| **MUTATE** | MCC | **0.811** | **0.922** | **0.784** | **0.964** | **0.885** | 0.501 |
| | $\mathcal{L}_{vae}$ | 0.670 | 0.448 | 0.508 | **0.253** | 0.942 | 1.201 |

## F    RELATED WORK

**Causal disentanglement and learning time series.**    Although estimating and predicting time series is a classical problem in both traditional statistics and modern machine learning, representation learning has opened new avenues for leveraging latent information to better characterize time series data (Wu et al., 2021;

Liu et al., 2024). Recently, learning causal representations in time series has become a foundational approach for enabling new scientific discoveries. This line of research primarily focuses on establishing identifiability of causal latent variables by exploiting nonstationary data (Yao et al., 2022b;a) and modular distribution shifts (Song et al., 2023; Cai et al., 2024) with sparsity constraints (Song et al., 2024; Zhang et al., 2024) on the latent transition. Those works solve the identifiability problem of latent causal models by leveraging sufficient variability that can come from proper interventions or passive distribution shifts. Another line of research focuses on learning the underlying causal graph among latent variables.

**Learning causal influences in stochastic processes.** While learning causality remains a considerably more challenging task than causal discovery or representation learning, several efforts have been made to bridge these areas. Here, we review existing approaches that link causal learning with stochastic modeling. Our scope is not limited to causal representation learning with stochastic processes, but extends to a broader set of problems that are closely related to either domain.

One representative direction in causal learning for dynamical systems is the study of Granger causality—a broader and looser notion compared to strictly structured causal models (Achab et al., 2018). It is widely acknowledged that full causal recovery in such systems is impossible. Consequently, even the most recent work on stochastic processes can only determine whether a point process $a$ is Granger-causal or non-causal with respect to another process $b$, typically formalized through *local independence* and the $\delta$-separation rule (Didelez, 2008). Another active line of work concerns identifiability in dynamical systems (Lippe et al., 2023). However, to the best of our knowledge, none of these models provides provable guarantees for highly dynamical systems such as self-exciting or more general stochastic processes.

Connections between causal representation and dynamical systems have also been explored through ordinary differential equations (ODEs) (Yao et al., 2024). Technically, these approaches recover only a set of parameters that are difficult to interpret as causal in the latent space, or at best allow stochastic dynamics in the observed variables. More recently, causal diffusion models have been proposed (Karimi Mamaghan et al., 2024; Lorch et al., 2024), yet they largely treat diffusion as a standard denoising process and thus do not permit a well-structured stochastic latent causal representation.

Another important research direction is investigate interventions on stochastic processes and the corresponding post-intervention distributions, which serve as the basis for causal inference (Sokol, 2013; Bongers et al., 2018; 2022; Boeken & Mooij, 2024; Lorch et al., 2024). The first attempt to introduce a causal interpretation into stochastic differential equations (SDEs) was made by authors of (Sokol & Hansen, 2014), where interventions are defined as the removal of single variables in SDEs. They showed that causal principles in SDEs can be formalized as interventions, with the resulting post-interventional distribution identifiable via the infinitesimal generator. However, such interventions are too restrictive to capture more complex dynamical scenarios. Following this initial line of work, (Bongers et al., 2018) further develops methods for estimating stationary causal models by minimizing the deviation of stationarity of diffusion. Nevertheless, they consider only observed diffusion processes that model causal effects from soft interventions that change the drifting term.

## G    EXTENDED DISCUSSION

CONNECTION TO THREE LATENT DYNAMIC PROCESSES

The identifiability guarantee is built on the proper weak convergence to finite-dimensional distributions, which reflects a two-way path between each pair of processes. We link them by the diagram of coupling and degeneration shown in A5.

DIFFERENT LEVEL IDENTIFICATION

Although causal representation learning is often regarded as resolved through identifiability guarantees, a deeper understanding of identifiability itself has been largely overlooked in the current literature. Here, we emphasize the distinction among different identifiability objectives, each presenting unique challenges, and argue that identifiability can be categorized into three major lines of research.

**Identifying $\mathcal{G}$.** Recovering the causal structure from observations is widely considered the most fundamental goal in causal representation learning—the very task that gave the field its name. Identifying $\mathcal{G}$ is central, as knowledge of the causal structure is often sufficient to uncover the underlying mechanisms, particularly for predicting post-intervention distributions. Given a set of representations for causal variables, the causal structure is fully recovered if the associated conditional independence constraints are uniquely determined. From a modeling perspective, the encoder outputs the distribution of a random variable $\hat{Z}$, which may not coincide exactly with the true causal variable $Z$. Nevertheless, this potentially "misaligned" representation still induces a valid causal structure, providing a principled way to decompose the observed distribution. Research along this direction is commonly referred to as latent causal structure learning (Jiang & Aragam, 2023; Jin & Syrgkanis, 2023; Zhang et al., 2023).

**Identifying latent $Z$.** This objective is more ambitious than merely recovering the latent structure $\mathcal{G}$, as it requires an exact component-wise correspondence for each causal variable. In general, the assumptions necessary for identifying the full latent variables tend to be stronger and less realistic. To recover $Z$, we assume that the unknown mixing function $f$ is noiseless and diffeomorphic. Leveraging information from the entire distribution allows us to guarantee an exact alignment between the estimated and true causal variables, thereby ensuring that the Jacobian of the construction map $f^{-1} \circ \hat{f}$ exhibits a sparse, permutation-like form. Most of prior works follow this line of research; we just name several representatives of them (Yao et al., 2022b; Zhang et al., 2024; Song et al., 2023; Hyvarinen & Morioka, 2016)

**Identifying full parameters and mixing $f$.** Finally, we conclude this section by comparing parameter-level identification, the most challenging one, to the two aforementioned goals. Note that identifying only $\mathcal{G}$ or $Z$, under some reduced conditions, could overlap with full parameter identification since $\mathcal{G}$ must be obtained if all causal parameters are identified. If the mixing is invertible or injective, one can easily recover latent variables by simply recovering the mixing function $f$ from the observation. However, when nothing is linear in both latent causal models and the mixing function, recovery of parameters means recovery of the entire generative model, which is an exacting task since the parameter space lives in an arbitrarily large ambient space but is not equipped with any closed form.

DISCUSSION ON NONPARAMETRIC INTENSITY

For completeness in identifiability theory, we complement our Theorem 2 and Theorem 3 with a discussion of a stochastic process featured by a nonlinear, flexible conditional intensity. Note that for real-world applications, Theorem 2 and Proposition 3 suffice. Following our reasoning, we can reformulate the nonlinear intensity process as follows:

$$O_t = f(Z_t^\Delta) \tag{G.1}$$

$$Z_t^\Delta = \psi(\lambda_t) + \epsilon_t \tag{G.2}$$

One may argue that it bears resemblance to the nonlinear time series process mostly addressed in Song et al. (2023); Yao et al. (2022b;a) , such a model $z_t = f(\text{Pa}(z_t), z_{t-1,j}) + \epsilon_t$. A conjecture is that the nonlinear mixing of linear intensity may or may not override the influence of the drastically increasing $\mathcal{F}_t$ up to the current sequence. Also, the intensity may be defined with an arbitrarily dense interval, which reflects the

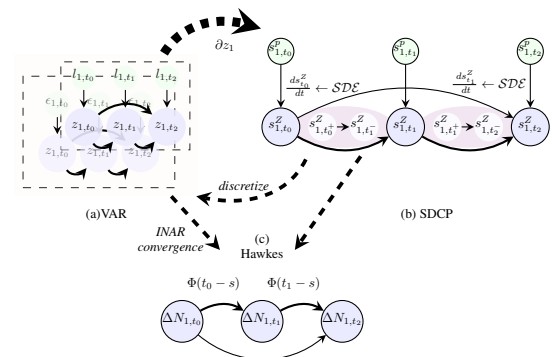

Figure A5: Connect three causal processes: revolution and degration of causal process.
(a) classic autoregressive model that allows time-delayed causal influences. (b) causal process featured by a stochastic differential dynamics. (c) Hawkes process, a special self-exciting process.NE
.

kernel effect. Therefore, we do not have the guarantee of strict identifiability for a rather long $\mathcal{F}_t$ -predictive process. Using the spectrum method can be a beneficial direction in future work.

## THE USE OF LARGE LANGUAGE MODELS

This paper uses LLM for an auxiliary purpose, including checking for typos and formatting. When some knowledge of a certain field cannot be accessed via a formal academic record (i.e., the publication does not have a trackable link, or the manuscripts contain handwriting rather difficult to parse), LLM is used just for such information acquisition.

