# OpenReview forum: "Causal Representation Meets Stochastic Modeling under Generic Geometry"
_ICLR.cc/2026/Conference — Submitted to ICLR 2026_

### Official Review · Reviewer_8Kot · 2025-10-31

**Soundness:** 3
**Presentation:** 2
**Contribution:** 2
**Rating:** 2
**Confidence:** 3

**Summary:**

The paper establishes novel connections between causal representation learning (CRL) and the modeling of latent stochastic processes in dynamical systems. Specifically, it provides identifiability results for stochastic processes that can be represented as INAR processes under a weak convergence assumption, showing when the latent causal structure is recoverable.
Building upon these results, the authors introduce a variational autoencoder framework (MUTATE), which features a time-adaptive transition module designed to learn causal latent variables in stochastic dynamic systems. The method is evaluated on synthetic datasets, comparing the identifiability of the recovered latent representations (measured via Mean Correlation Coefficient, MCC) against disentangled VAEs and prior temporal CRL approaches such as TDRL and PCL.

**Strengths:**

1. **Novel theoretical framing and mathematical soundness.** The connection between causal representation learning and continuous-time stochastic processes is original and addresses a relatively unexplored intersection between CRL and stochastic process theory. The identifiability results are grounded in solid algebraic and geometric reasoning, extending prior work on discrete-time latent causal processes.

2. **Clear theoretical motivation.** The use of weak convergence and INAR approximations provides a mathematically coherent bridge between continuous and discrete-time identifiability analysis.

**Weaknesses:**

1. **Limited Evaluation Metrics.** The empirical assessment is restricted to the MCC metric, which captures correlation-based identifiability but does not reflect disentanglement quality or causal graph recovery. Including other identifiability metrics (e.g. [3]) or explicit measures of causal edge recovery would strengthen the validation.

2. **Missing comparisons to recent CRL baselines.**
While the paper compares MUTATE against Beta-VAE, SlowVAE, and TDRL, it omits comparison with more recent temporal CRL methods, such as [1, 2], which also focus on identifiable causal dynamics.

3. **Synthetic-only “real-world” evaluation.** Although the authors mention testing on real-world biological data, the experiments rely on synthetic gene expression data generated by the SERGIO GRN simulator. This does not constitute a genuine real-world application and limits claims of practical relevance.

4. **Unclear experimental correspondence.**
It is not clearly stated which simulation settings correspond to Table 1 and Table A2, making it difficult to interpret the presented results. Further clarification of the data regimes and kernel/noise configurations would improve readability and reproducibility.

[1] "Causal Representation Learning for Instantaneous and Temporal Effects in Interactive Systems" Lippe et al. ICLR2023

[2] "Nonparametric Partial Disentanglement via Mechanism Sparsity: Sparse Actions, Interventions and Sparse Temporal Dependencies" Lachapelle et al. 2024

[3] "The Third Pillar of Causal Analysis? A Measurement Perspective on Causal Representations." Yao et al. NeurIPS2025

**Questions:**

see weaknesses

---

### Official Review · Reviewer_Tcwq · 2025-11-02

**Soundness:** 2
**Presentation:** 2
**Contribution:** 2
**Rating:** 6
**Confidence:** 3

**Summary:**

This paper investigates the problem of learning meaningful causal representations from observational data, with a particular focus on continuous-time latent stochastic point processes. It combines causal representation learning with stochastic modeling, providing theoretical identifiability guarantees and proposing a VAE-based solution framework called MUTATE. Experimental results demonstrate that MUTATE outperforms other methods.

**Strengths:**

1. The paper expands the scope of research on causal representation learning, extending it from traditional i.i.d. and discrete-time scenarios to continuous-time latent stochastic point processes.
2. The authors provide the theoretical analysis that the underlying causal structure and variables can still be uniquely identified under nonlinear mixing functions.

**Weaknesses:**

1. Although the theoretical framework relies on high-order cumulant tensor decomposition to achieve identifiability of causal representations, the practical solution shifts to a VAE-based generative modeling approach. This choice may lead to a disconnect between theory and practice, especially when the performance of the VAE depends on specific model assumptions or data distributions, potentially failing to fully reflect the advantages of the theoretical framework.
2. Theorem 2 relies on first-order Taylor expansion truncation and Jacobian linearization, but the authors did not provide an approximation error bound or a probabilistic statement regarding the rank lower bound of the Jacobian.
3. The experimental section did not include a comparison with the baseline method of directly estimating cumulants combined with CP decomposition.
4. What is the computational complexity of the proposed method in the paper? For example, the encoder-PSD flow decomposition involves Fast Fourier Transformation (FFT) at each step, which increases the overall complexity of the method.
5. Some statements in the paper are unclear. For instance, in the introduction, the sudden mention of "The cumulant propagates the causal structure through nonlinear transformations..." lacks sufficient explanation. What is the underlying insight behind this claim? How does the cumulant achieve the propagation of causal structure through nonlinear transformations?

**Questions:**

See the weaknesses above.

---

### Official Review · Reviewer_UzGW · 2025-11-04

**Soundness:** 3
**Presentation:** 2
**Contribution:** 3
**Rating:** 6
**Confidence:** 3

**Summary:**

This paper studies causal representation learning when the latent variables are continuous‑time multivariate point processes observed through an unknown non‑invertible mixing function. The author proves identifiability of the latent point process and its causal structure under linear mixtures (Thm. 1) and generic nonlinear mixing (Thm. 2), with necessary and sufficient conditions (Thm. 3). The author also proposes MUTATE, a VAE with a time‑adaptive transition and PSD‑whitening module to estimate the latent processes.

**Strengths:**

1. Most identifiability results in CRL assume discrete‑time latents and invertible mixing. Treating continuous‑time point processes with generic mixing is novel and practically relevant.
2. Although i did not look into the proof details, the theorems are intuitive and clear in a geometry view, considering identifibility as zero‑dimensionality of solution set of the system.
3. The model components are well related to the theorem.

**Weaknesses:**

Major Concerns

1. Only simulation results are presented. Considering the broad motivation of the work, including real dataset with event sequence from a continuous process would greatly strengthen the paper.
2. Adding a symbolic summary box would enhance readability, as the paper currently contains a large number of symbols such as \bigstar and \otimes.
3. Assumptions A2.2 and A3.2 are not clear to me, and under what circumstance should this assumption be true? It will be better to have some illustrations or examples. Besides, is the model robust against these assumptions?
4. The theory uses cumulants, and the algorithm instead uses a PSD whiteness prior and a time‑adaptive VAE without explicitly estimating cumulants. The connection is indirect.
5. As for as i know, to be self-exciting, the kernel should be postive and decreasing. This seems not contrained in the model. Could the author explain more on it?

Minor Concerns

6. Line 348: Hamilton conjugate -> Hermitian?
7. It will be better to have a figure or illustraion about the model pipeline to better demonstrate the model designation, and highlight the difference compared with other baseline models based on VAE.

**Questions:**

See Weakness.

---

### Official Review · Reviewer_vmtW · 2025-11-07

**Soundness:** 3
**Presentation:** 3
**Contribution:** 3
**Rating:** 6
**Confidence:** 2

**Summary:**

The paper tackles causal representation learning (CRL) for continuous-time latent point processes (Hawkes-type) under generic, potentially non-invertible mixing. It formalizes a weakly-convergent equivalence class to reconcile continuous-time dynamics with discrete observations and then proves identifiability (up to scaling/permutation or componentwise transforms) by analyzing the algebraic geometry of cumulant tensors and the associated ideals/varieties. On the algorithmic side, the authors propose MUTATE, a VAE with a time-adaptive transition and PSD-based prior decomposition to enforce independent noise and estimate latent dynamics. Simulations across several kernel families and a SERGIO-based gene-expression setting (all synthetic) show higher MCC than BetaVAE/SlowVAE/PCL/TDRL, with especially strong gains on exponential/power-law kernels (Table 1).

**Strengths:**

- Novelty: CRL for continuous-time latent Hawkes processes under generic (non-invertible) mixing is timely and underexplored; the weakly-convergent class neatly addresses discrete sampling vs continuous dynamics.
- The time-adaptive transition along with PSD decomposition to enforce latent whiteness provides connects theory to practice; the ELBO is explicitly provided.
- Empirical results show higher MCC on multiple kernel regimes vs temporal/non-temporal baselines.

**Weaknesses:**

- Assumption 2 (“zero-dimensional ideal) seems hard to verify.
- Lemmas 1–2 show convergence to a latent class but no explicit error rates are provided.

**Questions:**

The paper states an iff with Assumption 2 as necessary/sufficient for identifiability (Theorem 3), is there a practical test for the “linear degeneration” in Assumption 3(2)?

---

> ### Author Response · Authors · 2025-11-26
> **Response to vmtW (1)**
>
> Thanks for your valuable feedback on our work! We appreciate your recognition of our theoretical contribution, which is exactly the goal of this study. Since geometry can be challenging to follow, we acknowledge that some notations and theoretical concepts require further elaboration. In this rebuttal, we address your concerns by extending and demonstrating key parts of our theory.
>
> **Fundamental Condition for Identifiability**
> Our identifiability results stem from an intuitive idea: when a latent-space distribution is indexed by a finite number of parameters (in our setting the latent process $Z$ is determined by a continuous-convoluted kernel and Gaussian noise), identifiability amounts to cutting out a zero-dimensional variety from the associated system. To clarify this point, we added several demonstrations showing when different settings can or cannot satisfy our main theorem.
>
> *Single perfect intervention / variability in parameter space yields a zero-dimensional variety.*
> When the generic points $A_1, A_2, \ldots, A_p$, where $A = F (I - \Phi)^{-1}$, are uniquely determined up to scale and permutation, $FH$ is recovered as the unique set of generic points. Full identifiability then follows by ensuring that the associated variety
> $V : \langle F - K^{(k)} (I - \Phi) = 0,\ k = 1,2,\dots,p\rangle$
> has dimension zero. Geometrically, $\mathrm{dim}(V)$ matches the fundamental identifiability of the entire parameter space unless additional constraints are imposed. Applying Theorem 1.5 of [1] to both the upper and lower triangular parts of $\Phi$, identifiability is achieved using the minimal number of generic points $P = p$, matching the number of processes. This follows from considering the simpler ideal
> $\langle F - K^{(0)}(I-\Phi) - (F - K^{(k)}(I-\Phi))\rangle$,
> which degenerates into a linear system in a lower-dimensional ambient space.
>
> *Soft intervention on a single parameter per distribution cannot yield a zero-dimensional variety.*
> A single-node intervention or variability can recover the causal graph but cannot uniquely identify the full parameter under any benign ambiguity. We here lay out an illustrative process to explain why this happens: a soft intervention/variability do not make the corresponding row to be full zero, leading no generic/independent constraints introduced--therefore, the dimension of variety will not decrease.  For the full illustration, please see the revised section and appendix for proof.
>
> *Multiple interventions / variability*
> have a **positive** probability of reducing the parameter space to a zero-dimensional variety. The problem of identifying underlying causal graph and the variables for **General Linear Causal Representation** has been solved under *multiple intervention*[2]. However, how multiple intervention will cut out the full parameter space of $f$ and $\Phi$ is an open problem. We will add an illustration in the revised paper.
>
> Finally, identifiability of the entire finite-dimensional parameter space implies:
> 1. The exact latent causal processes or variables can be sampled from the parameter but are not explicitly recovered, since an encoder outputs a distribution.
> 2. Even when causal variables are uniquely recovered, the parameters do not need to match the “true” separate parameters exactly.
>
> The converse direction does hold: the map from $\Theta$ to $Z$ is not generally invertible, nor is the mixing function.
>
> [1] Paula Leyes Carreno, Chiara Meroni, and Anna Seigal. Linear causal disentanglement via higher-order cumulants. arXiv preprint arXiv:2407.04605, 2024.
>
> [2] Jin and Syrgkanis. Learning Causal Representations from General Environments Identifiability and Intrinsic Ambiguity,2023

---

### Meta-Review · Area_Chair_X5Xm · 2026-01-07

**Summary:**

Reviewers agree that this paper addresses a timely and underexplored problem - causal representation learning for continuous-time latent point processes under generic, potentially non-invertible mixing - and that the theoretical framing via algebraic geometry and identifiability is novel and mathematically sound. Strengths include the originality of extending CRL beyond discrete-time settings and the conceptual link between the identifiability theory and the proposed MUTATE framework. However,  reviewers also raise concerns about clarity and practicality, including strong assumptions that are difficult to verify, an indirect connection between cumulant-based theory and the VAE-based algorithm, and the absence of error bounds. On the empirical side, the evaluation is viewed as limited, relying exclusively on synthetic data and a narrow set of metrics, with missing comparisons.  Overall, while the work is interesting, I am recommending rejection due to theoretical opacity, limited empirical validation, limited author response, and overall clarity issues raised by multiple reviewers.

**Reviewer Concerns:**

The authors addressed some concerns in their rebuttal to one reviewer, but most concerns originally raised still remain.

**Reviewer Scores:**

I do not think that any of the reviewers would have changed their score, as authors only provided limited response (especially regarding the initial reject score).

---

### Decision · Program_Chairs · 2026-01-26

Reject